# Exploring the K+ binding site and its coupling to transport in the neurotransmitter:sodium symporter LeuT

**Solveig G Schmidt[1†], Andreas Nygaard[1†], Joseph A Mindell[2], Claus J Loland[1]\***

[1]Laboratory for Membrane Protein Dynamics, Department of Neuroscience, Faculty of Health and Medical Sciences, University of Copenhagen, Copenhagen, Denmark; [2]Membrane Transport Biophysics Section, National Institute of Neurological Disorders and Stroke, National Institutes of Health, Bethesda, United States

**Abstract** The neurotransmitter:sodium symporters (NSSs) are secondary active transporters that couple the reuptake of substrate to the symport of one or two sodium ions. One bound Na+ (Na1) contributes to the substrate binding, while the other Na+ (Na2) is thought to be involved in the conformational transition of the NSS. Two NSS members, the serotonin transporter (SERT) and the *Drosophila* dopamine transporter (dDAT), also couple substrate uptake to the antiport of K+ by a largely undefined mechanism. We have previously shown that the bacterial NSS homologue, LeuT, also binds K+, and could therefore serve as a model protein for the exploration of K+ binding in NSS proteins. Here, we characterize the impact of K+ on substrate affinity and transport as well as on LeuT conformational equilibrium states. Both radioligand binding assays and transition metal ion FRET (tmFRET) yielded similar K+ affinities for LeuT. K+ binding was specific and saturable. LeuT reconstituted into proteoliposomes showed that intra-vesicular K+ dose-dependently increased the transport velocity of [3H]alanine, whereas extra-vesicular K+ had no apparent effect. K+ binding induced a LeuT conformation distinct from the Na+- and substrate-bound conformation. Conservative mutations of the Na1 site residues affected the binding of Na+ and K+ to different degrees. The Na1 site mutation N27Q caused a >10-fold decrease in K+ affinity but at the same time a ~3-fold increase in Na+ affinity. Together, the results suggest that K+ binding to LeuT modulates substrate transport and that the K+ affinity and selectivity for LeuT is sensitive to mutations in the Na1 site, pointing toward the Na1 site as a candidate site for facilitating the interaction with K+ in some NSSs.

**\*For correspondence:**
cllo@sund.ku.dk

[†]These authors contributed equally to this work

**Competing interest:** The authors declare that no competing interests exist.

## eLife assessment

The bacterial neurotransmitter:sodium symporter homoglogue LeuT is a well-established model system for understanding the **fundamental** basis for how human monoamine transporters, such as those for dopamine and serotonin, couple ions with neurotransmitter uptake. Here the authors provide **convincing** data to show that K+ binding on the intraceullular side catalyses the return step of the transport cycle in LeuT by binding to one of the two sodium sites. The mechansitic consequences of K+ binding could either facilitate LeuT re-setting and/or prevent the rebinding and possible efflux of Na+ and substrate.

## Introduction

The family of neurotransmitter:sodium symporters (NSSs) include the transporters responsible for the reuptake of neurotransmitters from the extracellular space following synaptic transmission. Of pronounced interest to neuropharmacology is the subclass of monoamine transporters (MATs),

including the dopamine transporter (DAT), the norepinephrine transporter (NET), and the serotonin transporter (SERT), which are molecular targets for a range of psychopharmaceuticals, including drugs against depression, anxiety, neuropathic pain, attention-deficit hyperactivity disorder (ADHD), and narcolepsy (*Kristensen et al., 2011*; *Aggarwal and Mortensen, 2017*). They are also targets for psychostimulant drugs of abuse, such as cocaine and amphetamine (*Jayanthi and Ramamoorthy, 2005*; *Di Giovanni et al., 2016*). Thus, understanding the molecular basis underlying their ligand binding and transport are of physiological and pharmacological interest.

The NSS member LeuT, a hydrophobic amino acid transporter originating from the thermophile bacterium *Aquifex aeolicus*, was the first NSS for which the structure solved (*Yamashita et al., 2005*). LeuT has served as a structural and mechanistic model for NSS proteins (*Yamashita et al., 2005*; *Loland, 2015*). All NSS proteins are thought to share the LeuT-fold structure, comprising 10 transmembrane (TM) segments ordered in a pseudo-symmetry in the plane of the membrane between the first and the second five TMs. The LeuT-fold is mostly followed by two C-terminal TMs. Comparing the available structures stabilized in different states of the transport cycle suggests an overall similar transport mechanism, substantiating LeuT as a valid model protein (*Krishnamurthy and Gouaux, 2012*; *Wang et al., 2015*; *Yang and Gouaux, 2021*; *Shahsavar et al., 2021*; *Motiwala et al., 2022*). The substrate binding site is located halfway through the core of the protein and consists of coordinating residues from TM1, -3, -6, and -8 (*Yamashita et al., 2005*). It is flanked by two $Na^+$ binding sites, Na1 and Na2. The location and the residues forming the Na1 and Na2 sites are quite conserved between LeuT and the MATs. In the Na1 site, only one of the four coordinating residues differ by a serine to threonine substitution. In the Na2 site, two of the five residues are substituted between LeuT and MATs. A conserved feature shaping the $Na^+$ binding sites is the helical unwinding in TM1 and TM6, which exposes backbone-carbonyl oxygen atoms to partake in the ion coordination (*Yamashita et al., 2005*). In LeuT, the Na1 ion is also coordinated by the carboxyl-group of the substrate, yielding substrate binding highly $Na^+$-dependent (*Yamashita et al., 2005*).

The sodium gradient across the cell membrane is essential for driving substrate uptake in NSS proteins through occupation of the $Na^+$ sites. In addition, it has for long been recognized that SERT antiports $K^+$ as studies have shown that serotonin uptake was accelerated by an outward-directed $K^+$ gradient (*Nelson and Rudnick, 1979*). Consequently, $K^+$ antiport was suggested to increase the rate of the return step, which is thought to be the rate-limiting step of the transport cycle (*Hasenhuetl et al., 2016*; *Fitzgerald et al., 2019*). Cryo-EM reconstruction of SERT obtained in KCl revealed an inward-open conformation, suggesting this to be the most prevalent structure with KCl (*Yang and Gouaux, 2021*). The resolution of this structure, however, did not allow unambiguous identification of densities for bound ions. Thus, while the conformational details of the $K^+$-bound state of SERT are emerging, the location of the $K^+$ binding site remains unknown and the antiport mechanism not fully understood. We have shown that $K^+$ is antiported also by DAT from *Drosophila melanogaster* (dDAT) by a mechanism that shares similarities with $K^+$ antiport in SERT (*Schmidt et al., 2022*).

Previously, we have reported that LeuT interacts with $K^+$ and that the binding favors an outward-closed conformation (*Billesbølle et al., 2016*). Intra-vesicular $K^+$ also increased the concentrative capacity of [³H]alanine by LeuT, possibly by decreasing substrate efflux, but the details of $K^+$'s interaction with LeuT were not fully explored (*Billesbølle et al., 2016*). Here, we use purified LeuT, either in detergent micelles or reconstituted into proteoliposomes, to characterize the molecular components of how $K^+$ binds to LeuT, and how it influences the kinetics of substrate transport and the underlying conformational dynamics. Moreover, we mutate Na1 site residues and observe the effect on $Na^+$ and $K^+$ binding. Analogous to how LeuT for long has served as a structural model for the NSS family of proteins, it here serves as a functional model to suggest a role for $K^+$ in the transport mechanism.

## Results

We have previously shown that $K^+$ inhibits $Na^+$-dependent [³H]leucine binding to LeuT and changes its conformational equilibrium (*Billesbølle et al., 2016*). However, fundamental questions remain to be addressed: Is the effect of $K^+$ the result of direct binding to a specific binding site in LeuT? If so, where is this site located? What are the kinetic mechanisms underlying the effect of $K^+$ on [³H]alanine transport? To address this, we expressed His-tagged LeuT in *Escherichia coli*, harvested and solubilized the membranes in *n*-dodecyl-β-D-maltoside (DDM) and purified the protein by immobilized metal affinity chromatography. Purified LeuT was then used for radioligand binding assays, to analyze the

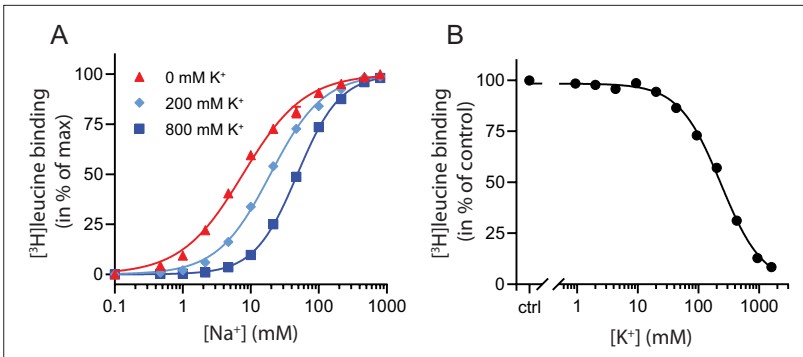

**Figure 1.** K$^+$ competitively inhibits Na$^+$-mediated [$^3$H]leucine binding in LeuT. (**A**) Na$^+$-mediated [$^3$H]leucine (10×
$K_d$) binding to purified LeuT in the absence (red triangles) or presence of 200 mM (light blue diamond) or 800 mM
(blue squares) K$^+$. Data are normalized to $B_{max}$ and fitted to a Hill equation, yielding EC$_{50}$ values in 0, 200, and
800 mM K$^+$ of 7.7 [7.3; 8.1], 19.4 [19.1; 19.8], and 48.5 [47.5; 49.5] mM, respectively. (**B**) K$^+$-dependent displacement
of Na$^+$-mediated [$^3$H]leucine binding in the presence of 7.7 mM Na$^+$, which corresponds to the EC$_{50}$ determined
in (**A**). Data are normalized to a control with 0 mM K$^+$ and fitted to a Hill equation with an IC$_{50}$ value of 234.8
[224.8; 243.1] mM. All data points are shown as mean ± standard error of the mean (s.e.m.), n=3–6 conducted in
triplicates. Error bars often smaller than data points. EC$_{50}$ and IC$_{50}$ values are reported as mean [s.e.m. interval]. The
ionic strength was maintained by substituting Na$^+$ and K$^+$ with Ch$^+$. All data is provided in the source data file.

The online version of this article includes the following source data and figure supplement(s) for figure 1:

**Source data 1.** Excel file containing data for (**A**) Na$^+$-mediated [$^3$H]leucine binding to purified LeuT in the absence
of K$^+$, and (**B**) K$^+$-dependent displacement of Na$^+$-mediated [$^3$H]leucine binding.

**Figure supplement 1.** K$^+$ inhibition is preserved for LeuT in low ionic strength.

**Figure supplement 1—source data 1.** Excel file containing data for displacement of Na$^+$-mediated [$^3$H]leucine
binding to LeuT by K$^+$, using Ch$^+$ or *N*-methyl-D-glucamine (NMDG$^+$) as the counter ion.

consequences of ion binding on conformational dynamics by transition metal ion FRET (tmFRET), as
well as for reconstitution into proteoliposomes for transport assays.

## K$^+$ binding is competing with Na$^+$ and saturable

To characterize the relationship between Na$^+$ and K$^+$ binding to LeuT, we investigated the effect of K$^+$
on Na$^+$-dependent [$^3$H]leucine binding using the scintillation proximity assay (SPA) (*Quick and Javitch,
2007*). Accordingly, we performed a Na$^+$ titration with 100 nM [$^3$H]leucine (*Figure 1A*). Choline (Ch$^+$),
seemingly inert to LeuT (*Billesbølle et al., 2016*), was used as counter ion to maintain the ionic
strength. Na$^+$ promoted the binding of [$^3$H]leucine to LeuT with an EC$_{50}$ for Na$^+$ of 7.7 mM, in line with
previous studies (*Zhao et al., 2011*; *Shi et al., 2008*). In the presence of K$^+$, however, the EC$_{50}$ right-
shifted while the $B_{max}$ remained unchanged, consistent with a competitive mechanism of inhibition
between Na$^+$ and K$^+$ as suggested previously (*Billesbølle et al., 2016*).

To determine whether the inhibition of [$^3$H]leucine binding by K$^+$ was saturable, we titrated in K$^+$
in the presence of Na$^+$ and [$^3$H]leucine. Displacement of [$^3$H]leucine binding by K$^+$ is concentration-
dependent, with an IC$_{50}$ for K$^+$ of 235 [225; 243] mM (mean [s.e.m. interval]) (*Figure 1B*). To ensure that
the displacement of [$^3$H]leucine by K$^+$ was not caused by artifacts originating from the high total salt
concentrations (1.6 M), we repeated the displacement assay with a total ionic strength of 208 mM salt
using either Ch$^+$ or *N*-methyl-D-glucamine (NMDG$^+$), both of which are inert to LeuT, as counter ions.
Again, K$^+$ dose-dependently displaced [$^3$H]leucine binding with an inhibition constant not significantly
different from that measured in the high salt conditions (*Figure 1—figure supplement 1*). Although
the approach is indirect with respect to K$^+$, this saturable, ionic strength-independent displacement of
[$^3$H]leucine binding is indicative of a specific, low affinity K$^+$ binding site in LeuT.

## The binding of ions is reflected in changes in the conformational equilibrium of LeuT

To obtain a more direct readout of K$^+$ binding to LeuT, we turned to tmFRET. This method relies on
the distance-dependent quenching of a cysteine-conjugated fluorophore (FRET donor) by a transition

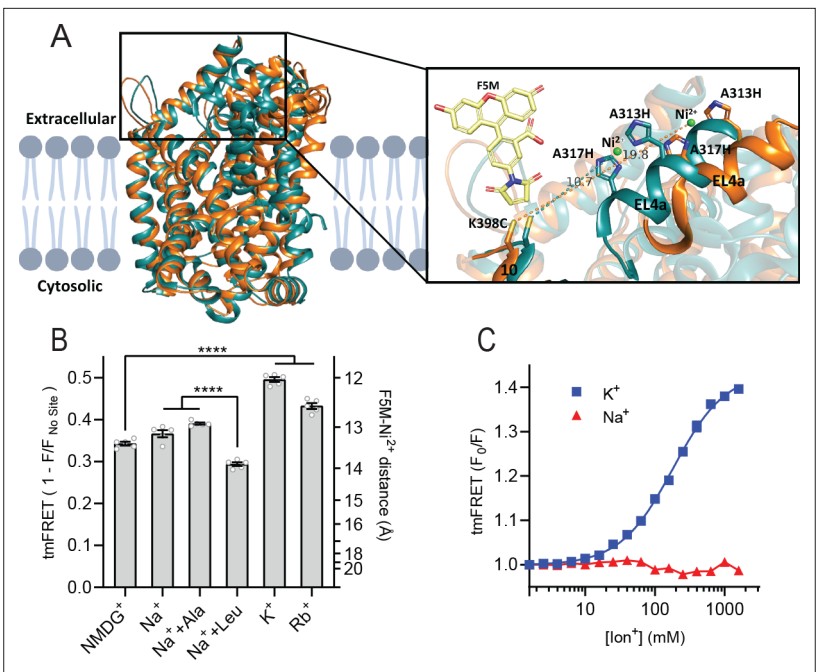

**Figure 2.** K+ shifts the conformational equilibrium toward an outward-closed state. (**A**) Left: cartoon representation of the superimposed structures of LeuT in the outward-open Na+-bound state (orange; PDB-ID 3TT1) and inward-open state (turquoise; PDB-ID 3TT3), viewed parallel to the plane of the membrane. Right: enlarged view of the top of transmembrane (TM)10 and extracellular loop (EL)4a, carrying the fluorescein-5-maleimide (F5M) (yellow, shown adjacent to K398C) and the Ni$^{2+}$-chelating His-X$_3$-His site (A313H-A317H). Dashed lines indicate for both structures the distance between the coordinated Ni$^{2+}$ ion (green) and the side chain sulfur of K398C (10.7 Å in 3TT3 and 19.8 Å in 3TT1). (**B**) Transition metal ion FRET (tmFRET) efficiencies (1 − $F/F_{no site}$) obtained for LeuT in 800 mM of the ions specified (±50 µM leucine or 200 µM alanine when indicated) upon saturation of the His-X$_3$-His site with 10 mM Ni$^{2+}$. Right vertical axis shows FRET efficiencies converted to distances by the FRET equation. ****p<0.0001 represents the significance levels from a Tukey multiple comparison-corrected one-way analysis of variance (ANOVA). (**C**) tmFRET ($F_0/F$) as a function of Na+ (red triangles) or K+ (blue squares) performed on LeuT$^{tmFRET}$ in the presence of 750 µM Ni$^{2+}$. The K+ response fitted to a Hill equation yields a Hill slope of 1.15±0.03 (mean ± s.e.m.) and an EC$_{50}$ of 182.6 [176.4; 189.1] mM, mean [s.e.m. interval]. All data points represent mean ± s.e.m. (error bars often smaller than data points), n=3–5 conducted in triplicates. All data is provided in the source data file.

The online version of this article includes the following source data and figure supplement(s) for figure 2:

**Source data 1.** Excel file containing data for *Figure 2B and C* providing transition metal ion FRET (tmFRET) efficiencies obtained for by the added ions in 10 mM Ni$^{2+}$.

**Figure supplement 1.** Transition metal ion FRET (tmFRET) principle and characterization of LeuT$^{tmFRET}$.

**Figure supplement 1—source data 1.** Excel file containing data for transition metal ion FRET (tmFRET) efficiencies as (B) a function of increasing Ni$^{2+}$ in ions, (C) for Rb+, and (D) background FRET signal.

metal (FRET acceptor), here Ni$^{2+}$, coordinated to an engineered α-helical His-X$_3$-His site (*Taraska et al., 2009a*). In LeuT, we have inserted the His-X$_3$-His site in extracellular loop (EL)4a (A313H-A317H) and a cysteine at the top of TM10 (K398C) that is labeled with fluorescein-5-maleimide (F5M). The distance between these FRET probes changes upon opening and closing of the extracellular gate in LeuT (*Figure 2A* and *Figure 2—figure supplement 1A*). Importantly, this construct, LeuT A313H-A317H-K398F5M (from here and on named LeuT$^{tmFRET}$), retains WT activity with respect to ligand binding affinities (*Billesbølle et al., 2016*). Accordingly, changes in tmFRET intensity is a conformational readout for both ion and ligand binding.

To determine the Ni$^{2+}$ concentrations required to saturate the His-X$_3$-His site, we first recorded the FRET efficiency as a function of increasing Ni$^{2+}$ for LeuT incubated with NMDG+, K+, or Na+± leucine (*Figure 2—figure supplement 1B*). To ensure close to saturating conditions for both Na+ and K+ while preserving the ionic strength, we applied 800 mM of the ions. The FRET efficiencies at saturating Ni$^{2+}$ concentrations reflect the average distance between the FRET probes and showed that K+ uniquely

stabilizes a high FRET state, suggesting a shift in the conformational equilibrium of the transporter toward an outward-closed state by K+ (*Figure 2—figure supplement 1A and B*). The Ni$^{2+}$ affinity for the His-X$_3$-His site is increased ~3-fold when Na+ or K+ is added to LeuT relative to NMDG+. This indicates that both Na+ and K+ bind and stabilize LeuT, including the His-X$_3$-His motif, whereas NMDG+ does not.

We proceeded by measuring the FRET efficiency only at saturating Ni$^{2+}$ concentrations (10 mM) to obtain FRET efficiencies independent of potential differences in Ni$^{2+}$ affinities. In addition to the conditions above, we measured FRET efficiencies for LeuT incubated in Na+ with alanine and in Rb+ (*Figure 2B*). Interestingly, applying Rb+, a cation often seen to be able to substitute for K+ in biological systems, also stabilized a more outward-closed state relative to that in NMDG+. The FRET efficiency in the Na+/leucine-bound state was decreased relative to that in the Na+-bound state, suggesting a, on average, more open-to-out state when adding leucine. We speculate that LeuT adopts a more conformationally restricted equilibrium upon binding of leucine relative to that with Na+ alone, and that this is reflected in a lower FRET efficiency (*Figure 2—figure supplement 1A*). This is also in line with the observation that alanine gives rise to a higher FRET efficiency than leucine (*Figure 2B*) as alanine binding allows a higher degree of conformational freedom in the transporter (*Zhao et al., 2011*; *Calugareanu et al., 2022*).

The large difference in tmFRET efficiencies between the apo state with NMDG+ and the K+-bound state of LeuT$^{tmFRET}$ allowed us to use the K+-induced change in conformational dynamics as a proxy for K+ binding. To estimate the affinity for K+ to apo state LeuT, we recorded the FRET efficiencies for LeuT$^{tmFRET}$ incubated with Ni$^{2+}$ in increasing concentrations of K+ (*Figure 2C*). We observed an increase in FRET as a function of K+ yielding an EC$_{50}$ of 183 [176; 189] mM, in line with the affinity determined by K+-dependent displacement of Na+ and [$^3$H]leucine (*Figure 1B*). This EC$_{50}$ is also in line with the affinity from the Schild analysis performed previously (*Billesbølle et al., 2016*). While Na+ did not impose major conformational changes, we found that also Rb+ induced a conformational response resembling that of K+, suggesting that Rb+ can indeed substitute for K+ although with a lower apparent affinity (*Figure 2—figure supplement 1C*). Of note, the high salt concentrations did not affect the intrinsic fluorescence properties of the fluorophore, validating that the responses to titration of the ions were direct results of conformational changes (*Figure 2—figure supplement 1D*). Along with the effect of K+ on Na+-dependent [$^3$H]leucine binding, this finding supports the existence of a specific K+ binding site in LeuT, and that K+ binding to this site induces an outward-closed conformation.

## K+ increases the rate of substrate uptake by LeuT

We have previously shown for LeuT reconstituted into liposomes that intra-vesicular K+ increases the concentrative capacity of [$^3$H]alanine, probably by decreasing its efflux (*Billesbølle et al., 2016*). To expand on these findings and to characterize how substrate transport was affected by the cations in the intra-vesicular buffer, we reconstituted purified LeuT into liposomes containing either Na+, NMDG+, Cs+, Rb+, or K+ (*Figure 3A*). Under each of these conditions, we measured time resolved [$^3$H]alanine uptake driven by a Na+ gradient (*Figure 3B*). With equimolar intra- and extra-vesicular Na+ concentrations, no [$^3$H]alanine transport was observed, indicating that the established Na+ gradient did drive [$^3$H]alanine uptake (*Figure 3B*). Interestingly, proteoliposomes containing K+ displayed a 2.5-fold increase in concentrative capacity compared to those containing Cs+ or NMDG+ (*Figure 3—figure supplement 1*; *Table 1*). To ensure that variations in the amount of active LeuT in the proteoliposomes did not affect the uptake capacity, we correlated it to the relative number of active LeuT under each condition (*Figure 3—figure supplement 1A*). The concentrative capacity with Rb+ was similar to that with K+, indicating that Rb+ can functionally substitute for K+. These data suggest that K+, and Rb+, are not obligate for LeuT transport, but add a concentrative potential for accumulation of alanine in addition to that obtained solely by the Na+ gradient.

Next, we investigated how the identity of the intra-vesicular cation affected alanine uptake rate. The $K_m$ for [$^3$H]alanine in vesicles containing K+ was 1.8±0.4 µM, and it was not significantly different upon substitution of intra-vesicular K+ with NMDG+, Cs+, or Rb+ (*Figure 3C* and *Figure 3—source data 2*). The $V_{max}$ for [$^3$H]alanine uptake was not significantly different between NMDG+- and Cs+-containing proteoliposomes. In contrast, the $V_{max}$ increased 2.5- to 3-fold when NMDG+ was substituted with Rb+ or K+. The increased uptake rate could originate from increases in the rates of certain steps in the transport cycle, from a decrease in [$^3$H]alanine efflux, or a combination of both.

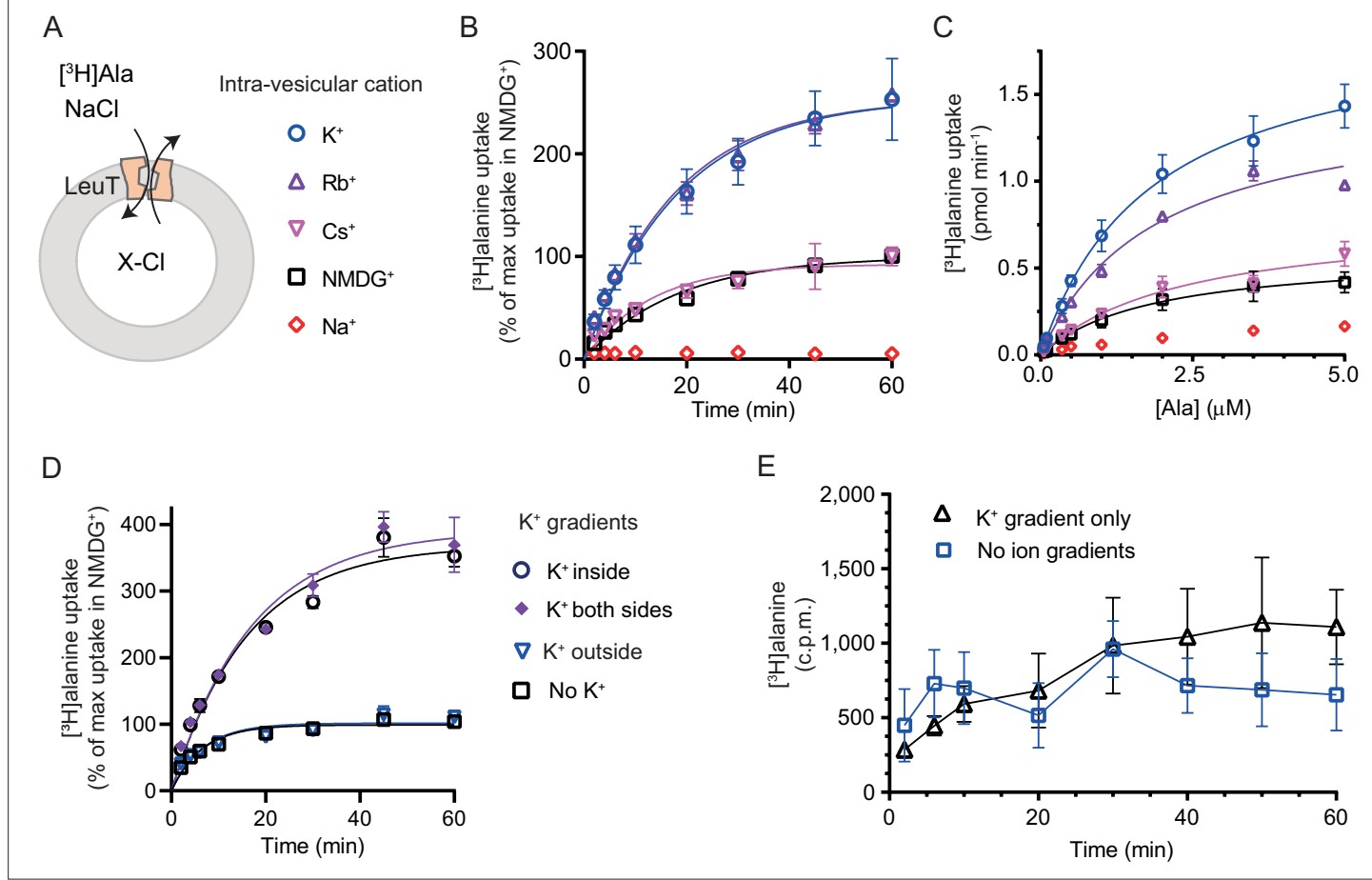

**Figure 3.** K+ modulates LeuT-mediated [3H]alanine transport. (**A**) Schematic of LeuT reconstituted into liposome containing buffer with various cations (**X**) and Cl- as corresponding anion. (**B**) Time-dependent [3H]alanine (2 µM) uptake in the presence of 200 mM Na+ into liposomes containing 200 mM of the various cations (colored as in (**A**)). Data are fitted to a non-linear regression fit. The maximum uptake predicted with N-methyl-D-glucamine (NMDG+) is defined as 100% (*Figure 3—source data 1*). (**C**) Concentration-dependent [3H]alanine uptake for 5 min (colored as in (**A**)). Lines are fit to Michaelis-Menten kinetics (*Figure 3—source data 2*). None of the $K_m$ values were significantly different (Tukey multiple comparison-corrected one-way analysis of variance [ANOVA], p>0.05). The $V_{max}$ value in Cs+ was not significantly different from NMDG+ (Tukey multiple comparison-corrected repeated-measures one-way ANOVA, p=0.5780), but K+ and Rb+ were significantly different from NMDG+ (p=0.0011 and 0.0132, respectively). (**D**) Time-dependent [3H]alanine uptake in the presence of 50 mM Na+ and absence of K+ (black squares) was defined as 100%. Addition of 150 mM K+ (blue triangles) did not significantly change the maximal uptake (unpaired t-test, p=0.92). With 150 mM intra-vesicular K+ (blue circles) the maximum uptake increased to 368±15% (mean ± s.e.m). This increase was not significantly (unpaired t-test, p=0.43) affected further when 150 mM K+ was also added to the extra-vesicular side (purple diamonds). The ionic strength was kept constant by substitution with NMDG+. (**E**) Time-dependent [3H]alanine uptake into liposomes containing 25 mM Na+ and 200 mM K+ in the presence of either 25 mM Na+ and 200 mM K+ (black line, no ion gradients) or 25 mM Na+ and 200 mM NMDG+ (blue line, outward-directed K+ gradient). The data points from the two conditions are not significantly different (unpaired Mann-Whitney test, p=0.645). All data points are shown as mean ± s.e.m (error bars), n=3 performed in duplicates (**B**) or triplicates (**C–E**). All data is provided in the source data file.

The online version of this article includes the following source data and figure supplement(s) for figure 3:

**Source data 1.** Word file containing data table for rate constants for time-dependent [3H]alanine transport by LeuT into PLs.

**Source data 2.** Word file containing data table for $V_{max}$ and $K_m$ of [3H]alanine transport by LeuT into PLs.

**Source data 3.** Word file containing data table for $V_{max}$ and $K_m$ of [3H]alanine transport as an effect by various K+ concentrations.

**Source data 4.** Excel file containing data for [3H]alanine uptake by LeuT reconstituted into PLs.

**Figure supplement 1.** [3H]alanine uptake into proteoliposomes.

**Figure supplement 1—source data 1.** Excel file containing data for [3H]alanine uptake into PLs with varying K+ or Na+ concentrations.

**Figure supplement 1—source data 2.** Files of original uncropped sodium-dodecyl-sulfate polyacrylamide gel electrophoresis (SDS-PAGE) gel shown in *Figure 3—figure supplement 1D*.

To gain further insight into the relationship between substrate transport and intra-vesicular K$^+$, we investigated how the K$^+$ concentration affected $K_m$ and transport velocity for [$^3$H]alanine. Accordingly, LeuT-containing proteoliposomes were prepared in a range of intra-vesicular K$^+$ concentrations. We observed that the $V_{max}$ for [$^3$H]alanine transport increased with increasing intra-vesicular K$^+$ concentration, whereas the estimated $K_m$ values for alanine were not significantly different (*Figure 3—figure supplement 1B* and *Figure 3—source data 3*), suggesting that the effect of K$^+$ on $V_{max}$ increases with increasing occupancy of the K$^+$ binding site. To assess how the concentration of extra-vesicular Na$^+$ affected the substrate uptake, we measured [$^3$H]alanine uptake at initial velocity conditions in increasing concentrations of extra-vesicular Na$^+$. We found that the half-saturating extra-vesicular Na$^+$ concentration was ~25 mM for vesicles containing 200 mM intra-vesicular K$^+$, which is in line with the higher affinity for Na$^+$ to LeuT compared to K$^+$ (*Figure 3—figure supplement 1C*).

The increase in [$^3$H]alanine transport rate by K$^+$ could be due to an imposed driving force by the outward-directed gradient. If so, dissipation of the K$^+$ gradient would decrease the [$^3$H]alanine transport rate. To allow for the addition of K$^+$ on the extracellular side also, we lowered the inward-directed Na$^+$ gradient to 50 mM. The lowered Na$^+$ gradient still resulted in an ~3.5-fold increased transport capacity with intra-vesicular K$^+$ relative to NMDG$^+$ (*Figure 3D*). However, dissipation of the K$^+$ gradient by application of equal amounts of K$^+$ on both sides did not change the transport capacity. Addition of K$^+$ only on the outside did also not change the transport capacity relative to that with the Na$^+$ gradient only (*Figure 3D*), and only having an outward-directed K$^+$ gradient could not drive [$^3$H]alanine transport (*Figure 3E*). These results suggest that it is the intra-vesicular K$^+$ per se that increases the transport rate of alanine and not a K$^+$ gradient.

It is difficult to control the directionality of proteins when they are reconstituted into lipid vesicles. They will be inserted in both orientations. Outside-out and inside-out. In the case of LeuT, it is the imposed Na$^+$ gradient which determines the directionality of transport. Uptake through the inside-out

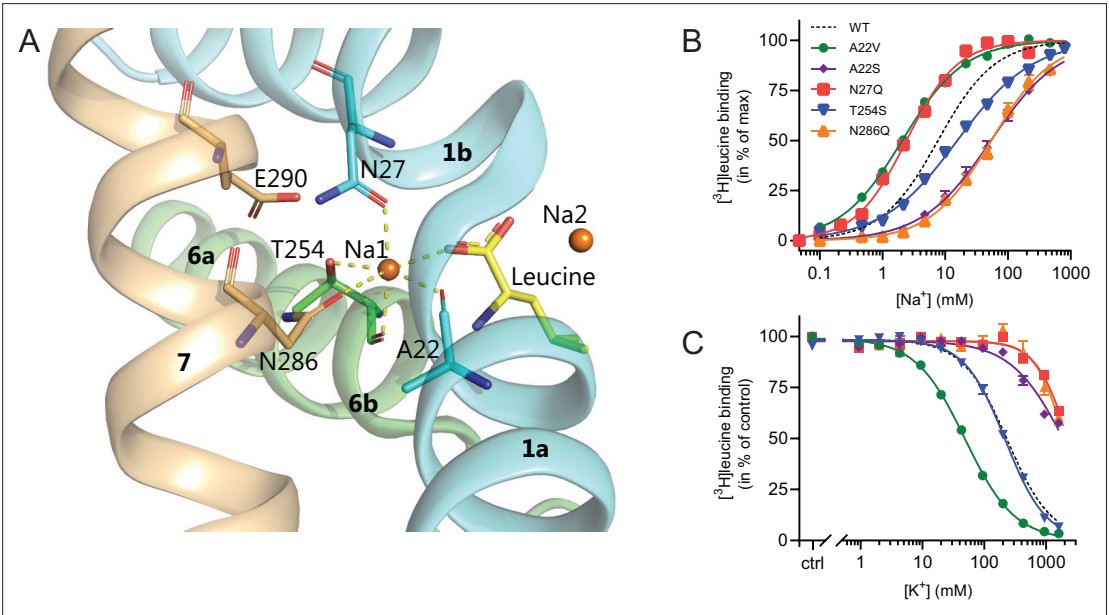

**Figure 4.** Na1 site mutants of LeuT display altered ion selectivity. (**A**) Cartoon representation of the Na1 site from the outward-occluded LeuT structure (PDB ID: 2A65) with the endogenous Na1 site coordinating residues along with the substrate leucine (yellow) and Glu290 labeled and shown as sticks. The coordination of Na$^+$ is depicted with dashed lines. Structural elements not involved in Na1 site are omitted for clarity. (**B**) Na$^+$-mediated [$^3$H]leucine (10× $K_d$) binding for Na1 site mutations, A22V (green circles), A22S (purple rhombi), N27Q (red squares), T254S (blue triangles), and N286Q (orange triangles), with WT shown for reference (black dashed line). Data are normalized to $B_{max}$ and fitted to a Hill equation. (**C**) K$^+$-dependent displacement of Na$^+$-mediated [$^3$H]leucine binding for the Na1 site mutants (colored as in (**B**)). Na$^+$ concentrations equivalent to the EC$_{50}$ values (determined in (**B**)) and 10× $K_d$ of [$^3$H]leucine for each mutant were used. Data points are normalized to a control without K$^+$ and modeled by the Hill equation. All data points are mean ± s.e.m., $n$=3–6. The ionic strength was maintained upon substitution with Ch$^+$. EC$_{50}$ and IC$_{50}$ values for (**B**) and (**C**) are summarized in *Table 2*. All data is provided in the source data file.

The online version of this article includes the following source data for figure 4:

**Source data 1.** Excel file containing data for Na$^+$-mediated [$^3$H]leucine binding for Na1 site mutations and how that is inhibited by K$^+$.

transporters will probably also happen. Note that the inside-out LeuT will not have the K[+] binding site exposed to the intra-vesicular environment. Accordingly, a propensity of transporters will likely not be influenced by the added K[+] and will tend to mask the contribution of K[+] on the transport mode from the right-side out LeuT. To investigate LeuT directionality in our reconstituted samples, we performed thrombin cleavage of accessible C-terminals on intact and perforated vesicles, respectively. The result suggests that the proportion of LeuT inserted as outside-out is larger than the proportion with an inside-out directionality (*Figure 3—figure supplement 1D*).

## Mutations in the Na1 site change the affinity for K[+]

With the elucidation of the impact of K[+] on LeuT substrate transport, we next sought to identify the binding site for K[+] in LeuT. Since K[+] and Na[+] binding are competitive and K[+] excludes substrate binding, we chose to focus on the Na1 site (*Figure 4*). Accordingly, we introduced the following conservative mutations of the amino acid residues in the Na1 site: A22S, A22V, N27Q, T254S, and N286Q. The aim was to keep LeuT functional but perturb K[+] binding. Since it has been shown that H[+] can substitute for the K[+] antiport in SERT (*Keyes and Rudnick, 1982*), we also mutated the adjacent Glu290, which has been proposed to facilitate H[+] antiport in LeuT (*Zomot et al., 2007*; *Forrest et al., 2007*; *Kantcheva et al., 2013*; *Malinauskaite et al., 2016*). Thus, by substituting it with glutamine (LeuT E290Q), we aimed to replicate the protonated state of Glu290. If the mechanism was similar to that in SERT but facilitated through this non-conserved residue, it should prevent K[+] binding.

All mutants were expressed in *E. coli* and purified in DDM. They all bound leucine and alanine (*Table 1*). The LeuT mutants A22V, A22S, and T254S retained WT-like substrate affinities whereas the mutation E290Q decreased the affinity one order of magnitude and the mutations N27Q and N286Q decreased the affinities about two orders of magnitude. To estimate their Na[+] affinities, we measured the Na[+]-dependent [[3]H]leucine binding in a next to saturating [[3]H]leucine concentration ($10\times K_d$), thereby taking the differences in substrate affinity for the individual mutants into account (*Figure 4B* and *Table 2*). Mutation of A22V and N27Q increased the Na[+] affinity by ~3-fold relative to WT. The T254S mutant caused a 2-fold decrease whereas the A22S and N286Q mutants both decreased the apparent Na[+] affinity by ~7-fold. The E290Q mutant retained close to WT Na[+] affinity.

To assess if the mutations in the Na1 site affected the ability to bind K[+] and test if the competitive mechanism of inhibition was preserved, we repeated the Na[+]-dependent [[3]H]leucine binding experiments in the presence of 800 mM K[+]. For LeuT WT, the added K[+] caused an ~6-fold increase in the $EC_{50}$ for Na[+]-dependent [[3]H]leucine binding. Interestingly, mutation E290Q and A22V resulted in increased antagonism by K[+] relative to WT, causing a 15-fold and 35-fold change, respectively (*Figure 5* and *Table 2*). For the remaining mutants, the antagonism by K[+] was either less (A22S, N27Q, and N286Q) or similar (T254S) (*Figure 5* and *Table 2*). The observation that the inhibition of Na[+]-dependent ligand binding by K[+] is retained for LeuT E290Q suggests that binding of K[+] is not dependent on a negative charge at Glu290 and can occur in parallel with H[+] antiport via Glu290 in LeuT. Also, the fact that the effect of K[+] on the T254S mutation was indifferent to WT suggests that the serine residue can completely substitute for threonine in this position.

Next, we assessed the impact of the mutations in the Na1 site on K[+] affinity by measuring the potency by which K[+] inhibits [[3]H]leucine binding. Again, [[3]H]leucine was added in a near-saturating concentration ($10\times K_d$) and Na[+] at its determined $EC_{50}$. We observed a marked (~10-fold) decrease in K[+] sensitivity for A22S, N27Q, and N286Q, suggesting perturbed K[+] binding by these mutants (*Figure 4C*, *Figure 5*). Note that the same N27Q mutant had an increased $EC_{50}$ for Na[+]. In contrast, the potency for K[+] in inhibiting [[3]H]leucine binding was almost 5-fold increased by the A22V mutation (*Figure 4C*, *Figure 5*, and *Table 2*). The T254S mutation did not alter the K[+] sensitivity relative to LeuT WT (*Figure 4C*, *Figure 5*, and *Table 2*). The cognate position to Thr254 in SERT is also a serine residue, which could indicate that the threonine to serine substitution is tolerated in terms of K[+] sensitivity between NSSs from different species.

Taken together, the Na1 site mutations showed a marked and differentiated response to the effects of both Na[+] and K[+]. For some mutants, the affinities for both Na[+] and K[+] were decreased (A22S and N286Q) or increased (A22V). For one, the effects were differentiated so that the Na[+] affinity was increased, while the K[+] affinity was decreased (N27Q).

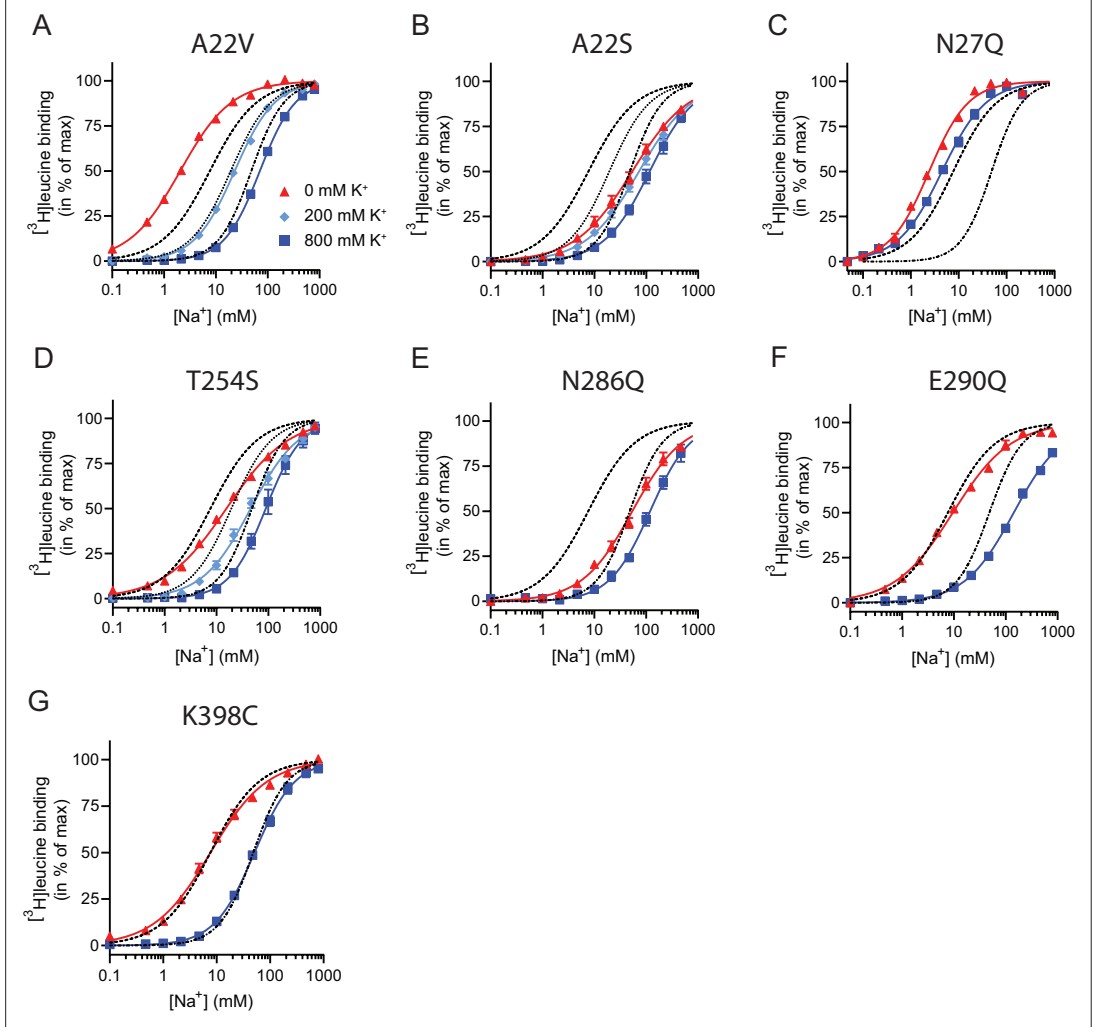

**Figure 5.** Na⁺-mediated [³H]leucine binding and inhibition by K⁺ by the LeuT mutants. (**A–F**) Na⁺-mediated [³H]leucine (10× $K_d$) binding for the Na1 site mutants performed in the absence (red triangles) and presence of 200 mM (light blue diamond) and 800 (blue squares) mM K⁺. (**A**) A22V, (**B**) A22S, (**C**) N27Q, (**D**) T254S, (**E**) N286Q, (**F**) E290Q, and (**G**) K398C. LeuT K398C was included as negative control. WT is shown for reference for 0 mM (dashed line), 200 mM (dotted line), and 800 mM (dash-dotted line) of K⁺. Ionic strengths were maintained using Ch⁺. All data points are shown as mean ± s.e.m. normalized to $B_{max}$ and fitted to a Hill model. EC₅₀ values are summarized in **Table 2**, n=3–4. All data is provided in the source data file.

The online version of this article includes the following source data for figure 5:

**Source data 1.** Excel file containing data for Na⁺-mediated [³H]leucine binding for the Na1 site mutants and in the presence of K⁺.

## The conformational responsiveness is altered in the Na1 site mutants

With coordinates from TM1, -7 and the bound substrate, the Na1 site is likely a central mediator of conformational changes. To investigate how the conformational equilibria in the transporters were affected by the Na1 site mutations, we introduced them into the LeuT$^{tmFRET}$ background and probed their response to Na⁺, K⁺, and substrate binding with respect to changes in tmFRET. We first investigated the conformational equilibria for N286Q$^{tmFRET}$ in NMDG⁺ (apo state), in Na⁺ with and without leucine, and in K⁺. Surprisingly, even though the EC₅₀ and IC₅₀ values for Na⁺ and K⁺, respectively, were markedly increased (decreased affinities), the conformational equilibria of N286Q resembled those of LeuT WT (**Figure 6A**). This suggests that the substantial decreases in ion affinities and selectivity, imposed by the asparagine to glutamine substitution, do not result from conformational biases, but likely from a direct mutual modulation of their binding site. However, we failed to observe any [³H]alanine transport activity by reconstituted LeuT N286Q, likely as a result of its low substrate affinity.

The N27Q$^{tmFRET}$ showed a markedly lower tmFRET efficiency in the apo state compared to LeuT-$^{tmFRET}$ (**Figure 6B**), suggesting that the mutation biases the conformational equilibrium toward a more

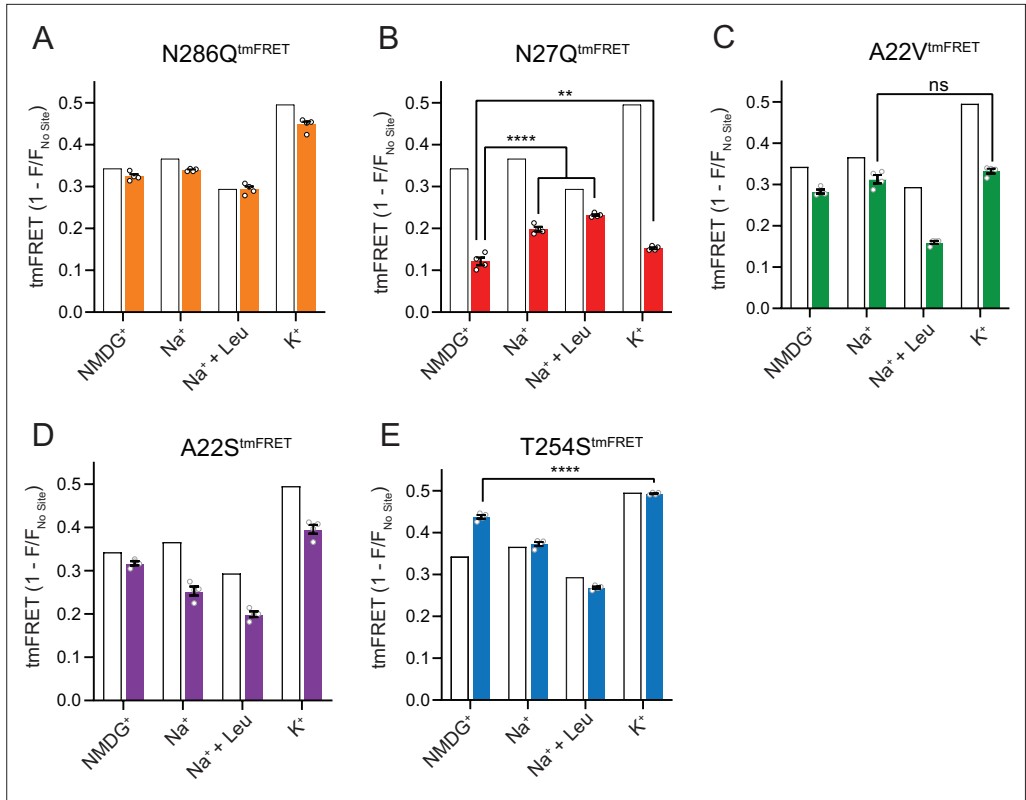

**Figure 6.** The Na1 site LeuT mutants exhibit different conformational equilibria. (A–E) Transition metal ion FRET (tmFRET) efficiencies (1 – $F/F_{no\ site}$) for N286Q[tmFRET] (orange, **A**), N27Q[tmFRET] (red, **B**), A22V[tmFRET] (green, **C**), A22S[tmFRET] (purple, **D**), and T254S[tmFRET] (blue, **E**) incubated in 800 mM of the indicated ions and 50 μM for leucine. The His-X$_3$-His site was saturated with 10 mM Ni$^{2+}$. The corresponding tmFRET efficiencies for LeuT[tmFRET] (white bars) are shown for reference. All data points are mean ± s.e.m., $n$=4, performed in triplicates. In (**B**), **p<0.001; ****p<0.0001 represent the significance levels from a Tukey multiple comparison-corrected one-way analysis of variance (ANOVA), comparing the mean obtained in K$^+$ with that in $N$-methyl-D-glucamine (NMDG$^+$), and the mean in Na$^+$ and Na$^+$±leucine with that in NMDG$^+$. In (**C**) and (**E**), ns, not significant and ****p<0.0001 using a one-way ANOVA with Bonferroni multiple comparison correction. All data is provided in the source data file.

The online version of this article includes the following source data and figure supplement(s) for figure 6:

**Source data 1.** Excel file containing data for transition metal ion FRET (tmFRET) efficiencies for all mutants in $N$-methyl-D-glucamine (NMDG$^+$), Na$^+$, leucine, or K$^+$.

**Figure supplement 1.** [³H]alanine uptake by selected Na1 mutants.

**Figure supplement 1—source data 1.** Excel file containing data for [³H]alanine transport activity over time by selected Na1 mutants.

outward-open conformation. Addition of Na$^+$ and leucine restored the conformational equilibrium toward significantly more outward-closed states, but not to the same extent as found for the LeuT[tmFRET] construct. However, we hardly observed any change in the tmFRET efficiency with K$^+$ in the N27Q mutant. This is in line with its marked decrease in K$^+$ affinity. The result could either suggest that Asn27 is important for the binding of K$^+$, or that the mutation causes a conformational bias which makes this mutant rarely visit the K$^+$-selective state. We were unable to measure any [³H]alanine transport activity by the LeuT N27Q mutant when reconstituted into K$^+$-containing vesicles.

Mutant A22V[tmFRET] displayed similar tmFRET efficiency as LeuT[tmFRET] in NMDG$^+$ and Na$^+$, but the tmFRET efficiency induced by K$^+$ was not significantly different from that induced by Na$^+$ (**Figure 6C**). This indicates that the conformational equilibrium of A22V[tmFRET] apo-form is not changed by the mutation. In addition, although both Na$^+$ and K$^+$ ions bind with a higher affinity to LeuT A22V than to WT (**Figure 4**), the tmFRET data suggest that K$^+$ no longer promotes the conformational shift toward the outward-closed conformation. To further evaluate this, we reconstituted the mutant into liposomes and measured its ability to transport alanine (**Figure 6—figure supplement 1A**). With equimolar

$Na^+$ on each side of the lipid bilayer, we observed a specific [³H]alanine signal, which could either reflect transport driven by the [³H]alanine gradient alone or simply binding of [³H]alanine to LeuT. In the presence of a $Na^+$ gradient, we observed a minor, but significant increase in specific [³H]alanine counts. However, the substitution to intra-vesicular $K^+$ did not affect the [³H]alanine activity. Further investigations must clarify whether the changes in observed [³H]alanine activity constitute a transport or a binding event.

The LeuT A22S mutant displayed a decrease in both $Na^+$ and $K^+$ affinity (***Figure 4***). When inserting it into the LeuT^tmFRET background (A22S^tmFRET), the pattern in tmFRET efficiencies were largely unaltered from LeuT^tmFRET, although with a minor reduction in responses upon addition of ligands (***Figure 6D***). This suggests that the mutation only has minor effect on the overall conformational equilibrium independent of the added ligands. When inserted into liposomes, LeuT A22S retained the ability to transport alanine in the presence of a $Na^+$ gradient. As for LeuT WT, intra-vesicular $K^+$ increased the concentrative capacity for [³H]alanine transport. However, the increase was 3-fold higher than what we observed for LeuT WT (***Figure 6—figure supplement 1B***). The WT-like conformational equilibrium could suggest that the decreased $Na^+$ and $K^+$ affinities are due to a direct perturbation of the ion binding site by the A22S mutation.

Finally, we characterized the tmFRET response for T254S^tmFRET. The mutant exhibited a slightly higher tmFRET efficiency in the apo state (NMDG^+), but its conformational response to $Na^+$, substrate, and $K^+$ binding was similar to LeuT^tmFRET (***Figure 6E***). Reconstituted into liposomes, LeuT T254S transported [³H]alanine and retained a WT-like increase in concentrative capacity with intra-vesicular $K^+$ (***Figure 6—figure supplement 1C***). As the substitution is the only apparent difference between the Na1 site in LeuT relative to the human SERT, DAT, and NET, it could suggest a similar functionality of the Na1 site by the four transporters.

In all, although we are unable to discern between direct and indirect effects imposed by the mutants, our results do reflect both concerted and opposed consequences of $Na^+$ and $K^+$ binding, conformation, and substrate transport.

## Discussion

In this study, we have examined the binding of $K^+$ ions to purified LeuT stabilized in detergent micelles. We have determined the binding potency through competition binding with $Na^+$ and radiolabeled ligand, and by changes in the conformational equilibrium of the transporter induced directly by $K^+$, as well as explored the role of $K^+$ in the transport process with LeuT reconstituted into liposomes. Additionally, we have investigated the binding site for $K^+$ by a mutational screen of the residues contributing to the already known sodium binding site, Na1, which is conserved among NSSs.

To define an interaction between a ligand and a protein as being the result of a binding site, it must be saturable. Here, we show that $K^+$ binding is saturable both when assessing its inhibition of $Na^+$-dependent [³H]leucine binding and when applying tmFRET as a direct conformational readout. The affinity is around 180 mM measured with tmFRET and by the Cheng-Prusoff equation, applied for competitive inhibitors, the $IC_{50}$ from the $Na^+$-dependent [³H]leucine binding converts to a $K_i$ of 124 [122; 127] mM. This is in the same range as reported previously (***Billesbølle et al., 2016***). Even though ions are quite abundant in many biological systems, an affinity above 100 mM is rather low. We can only speculate whether this is within a physiological range for *A. aeolicus*. However, even at low occupancy, e.g., at its $K_i$ when occupancy is 50%, the bound $K^+$ would regulate the transport velocity.

TmFRET provides a means for a direct measurement of $K^+$ binding to LeuT based on changes in the conformational equilibrium of the ensemble of transporters. The tmFRET efficiencies in this study reflects intramolecular distance changes between the extracellular side of TM10 and EL4. According to the solved LeuT crystal structures, it predicts that this distance will gradually decline upon transition from the outward-open, through the occluded, to the inward-open conformation (***Yamashita et al., 2005***; ***Krishnamurthy and Gouaux, 2012***). $K^+$ specifically induces a high FRET efficiency relative to those of the other ions and ligands tested, and titration with $K^+$ fits a Hill model with a slope of 1.15±0.03 (mean ± s.e.m.), which is in accordance with the existence of one $K^+$ binding site in LeuT that – when occupied – biases the conformational equilibrium towards an outward-closed state. Interestingly, $Rb^+$ induced a response similar to that of $K^+$, which correlates with previous observations that $Rb^+$ can substitute for $K^+$ binding due to similarities in size and preferred coordination (***Billesbølle et al., 2016***). The decreased FRET efficiency in the $Na^+$/leucine-bound state, relative to that in the

Na$^+$-bound state, could suggest that the binding of leucine on average favors a more open-to-out state, although this contradicts with the known crystal structures. However, we speculate that instead this result could be a consequence of the principle by which steady-state FRET efficiencies for heterogeneous dynamic ensembles are biased toward shorter distances (*Taraska et al., 2009a*; *Beechem and Haas, 1989*; *Lakowicz, 2006*). If LeuT adopts a more conformationally restricted equilibrium upon binding of Na$^+$ and leucine relative to that with Na$^+$ alone, thereby lowering the frequency by which the transporter visits the outward-closed state, this could reflect the lower FRET efficiency observed for the Na$^+$/leucine-bound state.

When we reconstituted LeuT into liposomes, intra-vesicular K$^+$ increased the capacity and $V_{max}$ for [$^3$H]alanine uptake compared to intra-vesicular NMDG$^+$ and Cs$^+$. Rb$^+$ displayed a similar effect, suggesting that the conformational effect of Rb$^+$ measured by tmFRET translates to an effect on transport function. Dissipating the K$^+$ gradient or adding K$^+$ solely on the extra-vesicular side of the proteoliposomes did not affect uptake capacity. Neither were a K$^+$ gradient alone able to drive uptake of [$^3$H]alanine. This suggests that the effect of K$^+$ on LeuT uptake is independent of a K$^+$ gradient. Conservative mutations in the Na1 site and the adjacent residue Glu290 retained the ability of the transporter to bind ligands, but only A22S and T254S showed sustained [$^3$H]alanine transport into proteoliposomes, and a persistent functional effect of K$^+$. Interestingly, the LeuT N27Q possessed low affinity for K$^+$ but increased Na$^+$ affinity. Also, the tmFRET data suggested it irresponsive to K$^+$, but also showed an altered conformational equilibrium. This could suggest that the N27Q mutation causes a bias toward an outward-open conformation. [$^3$H]Alanine transport by LeuT A22V showed no effect by the addition of intra-vesicular K$^+$. This correlates with the loss of apparent K$^+$-induced conformational response as assessed in the tmFRET studies (*Figure 6*). However, even though the mutant showed Na$^+$-dependent [$^3$H]alanine transport, the rate constant was high (1.55 min$^{-1}$) relative to LeuT WT (0.056 min$^{-1}$). This could suggest that the A22V mutation influences additional functional properties, such as entering an exchange mode. Further investigations are needed to clarify this.

Despite the known role of K$^+$ in the transport mechanism of SERT and other NSSs, only few studies have attempted to localize the K$^+$ binding site (*Billesbølle et al., 2016*; *Mari et al., 2004*; *Khelashvili et al., 2016*). Mutations of Asn338 (corresponding to Asn286 in LeuT) in the Na$^+$- and K$^+$-coupled amino acid transporter KAAT1 from *Manduca sexta* suggest that this residue is important for cation selectivity and coupling (*Mari et al., 2004*). Molecular dynamics simulations of LeuT showed that the K$^+$ ion could jump between the Na2 and Na1 sites (*Khelashvili et al., 2016*). Mutation in the Na2

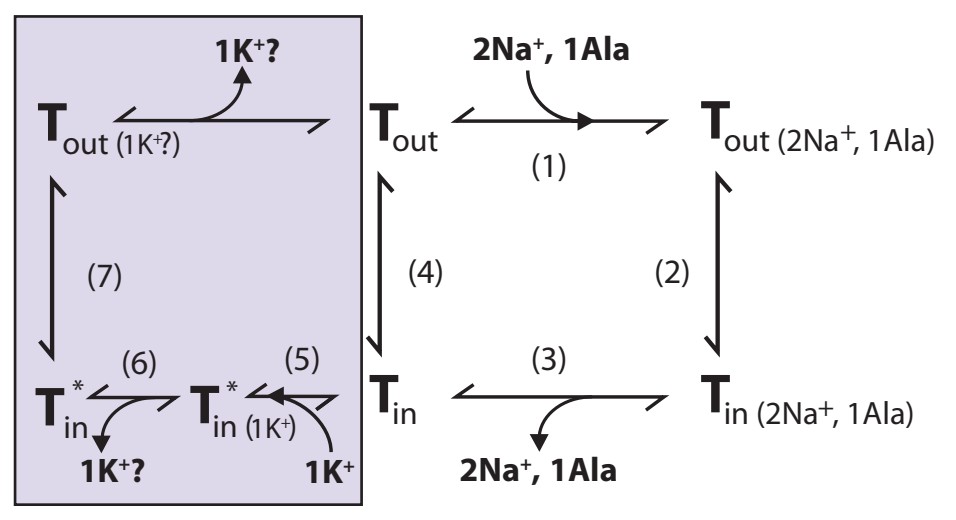

**Figure 7.** Proposed role of K$^+$ in the translocation cycle. Here, exemplified for alanine (Ala) as substrate. LeuT apo-form will likely reside in an equilibrium between its outward- ($T_{out}$) and inward-facing ($T_{in}$) conformation. After Na$^+$ and Ala are bound (1) and released (3), LeuT can either (i) rebind Na$^+$ and Ala, which will promote efflux (2); (ii) transition to the outward-open conformation in its apo-form (4), or (iii) bind K$^+$ (5). K$^+$ binding will promote an inward-facing LeuT state ($T^*$) which is unable to bind Na$^+$, either by competitive inhibition or by promoting a state that does not allow Na$^+$ binding. From here, K$^+$ could either be released again to the intracellular environment (6) or be counter-transported (7).

site of LeuT (T354V) resulted in a transporter locked in the outward-closed state, and mutation in the Na1 site (T254V) resulted in a transporter which showed a decreased conformational response to $K^+$ (*Billesbølle et al., 2016*).

To the best of our knowledge, this work represents the first systematical examination of $K^+$ binding in the Na1 site. We found mutations that either increase or decrease the affinities for both $Na^+$ and $K^+$, but we also saw mutations with opposite effects with respect to the affinities, and thereby selectivity, for the two ions. These changes in ion affinities were independent of conformational biases as assessed with tmFRET. This decoupling in ion selectivity and conformational bias suggests that the alterations result from direct modulations of the binding site for $K^+$. However, binding of the two $Na^+$ ions and substrate in LeuT is synergistic, making it difficult to completely isolate effects of mutations to a specific site. Consequently, we cannot exclude the possibility that $K^+$ binding could also involve the Na2 site or another unknown site. Nevertheless, we consider our mapping of the effect on $K^+$ binding by mutations in Na1 a robust starting point for the quest to identify the $K^+$ binding site in NSSs.

We propose that $K^+$ binding either facilitates LeuT transition from inward- to outward-facing (the rate-limiting step of the transport cycle) or solely prevents the rebinding and possible efflux of $Na^+$ and substrate. It could also be a combination of both. Either way, intracellular $K^+$ will lead to an increase in $V_{max}$ and concentrative capacity. Note that our previous experiment showed an increased [³H]alanine efflux when LeuT transports alanine in the absence of intra-vesicular $K^+$ (*Billesbølle et al., 2016*). Specifically, the mechanistic impact of $K^+$ could be to catalyze LeuT away from the state that allows the rebinding of $Na^+$ and substrate. This way, $K^+$ binding would decrease the possible rebinding of intracellularly released $Na^+$ and substrate, thereby rectifying the transport process and increase the concentrative capacity and $V_{max}$ (*Figure 7*). Our results suggest that $K^+$ is not counter-transported but rather promotes LeuT to overcome an internal rate-limiting energy barrier. However, further investigations must be performed before any conclusive statement can be made here. If $K^+$ is not counter-transported, LeuT might comply with the mechanism previously suggested for the human DAT (*Bhat et al., 2021*).

Taken together, $K^+$ binding seems conserved between LeuT, SERT (*Rudnick and Nelson, 1978*; *Schicker et al., 2012*), dDAT (*Schmidt et al., 2022*), and potentially human DAT (*Schmidt et al., 2022*; *Li and Reith, 2000*). Also, the ability of $K^+$ binding to induce a conformational change toward an outward-closed/inward-open state appears to be a common mechanism (*Yang and Gouaux, 2021*; *Schmidt et al., 2022*; *Schicker et al., 2012*; *Möller et al., 2019*). The conservation of $K^+$ binding, but lack of $K^+$ antiport, has also been observed in a bacterial member of the SLC1 carrier family (*Ryan et al., 2009*; *Wang et al., 2019*). Perhaps, regulation of $Na^+$-dependent substrate transport by $K^+$ is a more common mechanism than previously anticipated.

## Materials and methods

**Key resources table**

| Reagent type (species) or resource | Designation | Source or reference | Identifiers | Additional information |
|---|---|---|---|---|
| Gene (Aquifex aeolicus) | LeuT gene | https://doi.org/10.1038/ncomms12755 | | |
| Strain, strain background (*Escherichia coli*) | BL21(DE3) | Sigma-Aldrich | CMC0016 | Electrocompetent cells |
| Strain, strain protein expression (*Escherichia coli*) | C41(DE3) | Sigma-Aldrich | CMC0017 | Cells for protein expression |
| Recombinant DNA reagent | LeuT; LeuT WT (plasmid) | https://doi.org/10.1038/ncomms12755 | | pET16b backbone |
| Recombinant DNA reagent | LeuT^tmFRET; LeuT A313H-A317H-K398C (plasmid) | https://doi.org/10.1038/ncomms12755 | | pET16b backbone |
| Recombinant DNA reagent | LeuT^K398C; LeuT K398C (plasmid) | https://doi.org/10.1038/ncomms12755 | | pET16b backbone |

*Continued on next page*

*Continued*

| Reagent type (species) or resource | Designation | Source or reference | Identifiers | Additional information |
|---|---|---|---|---|
| Sequence-based reagent | Primers for generation of LeuT mutants | This paper | | See list of primers in Appendix. Primers were synthesized by Eurofins Genomics |
| Chemical compound, drug | fluorescein-5-maleimide; F5M | Thermo Fischer Scientific | F150 | |
| Chemical compound, drug | [³H]leucine | PerkinElmer | NET1166001 | |
| Chemical compound, drug | *E. coli* polar lipid extract | Avanti Polar Lipids | 100600 C | |
| Chemical compound, drug | [³H]alanine | Moravek Biochemicals | | Made on request |
| Chemical compound, drug | Gel stain InstantBlue | abcam | Ab119211 | |
| Software, algorithm | GraphPad Prism 9.0 | GraphPad Software, Boston, Massachusetts USA | | |
| Other | Yttrium Silicate Copper (YSi-Cu) His-tag SPA beads; YSi-Cu His-tag SPA beads | PerkinElmer | RPNQ0096 | SPA beads used for scintillation proximity assays - See: Pharmacological characterization of LeuT mutants. In Materials & Methods Section |
| Other | thrombin | Cytiva | 27084601 | Protease used to cleave the His-tag from LeuT |
| Other | Filtermat B – GF/B | PerkinElmer | 1450–521 | Filters used to trap proteoliposomes for scintillation counting of [3 H]ligand |
| Other | MeltiLex B/HS | PerkinElmer | 1450–442 | Scintillation plates to add on Filtermats for [3 H] counting |
| Other | Nuclepore Track-Etch Membrane polycarbonate filter of pore size 400 nm | Sigma-Aldrich | WHA10417104 | For extrusion of PLs. |
| Other | SM-2 Bio-Beads | Bio-Rad Laboratories | 1523920 | For detergent extraction from the specimen to promote reconstitution of LeuT into PLs |

## Reagents

Unless otherwise stated, reagents were purchased from Merck (Life Sciences).

## Cloning of LeuT mutants

DNA encoding LeuT from *A. aeolicus,* fused C-terminally to a thrombin protease cleavage site and an octahistidine tag, was cloned into a pET16b vector. Single mutations in the Na1 site (A22V, A22S, N27Q, T254S, N286Q, E290Q) were inserted into LeuT constructs with pre-existing pairs of mutations (A313H-A317H-K398C; K398C) and without (WT). Full gene sequences were verified by DNA sequencing (Eurofins Genomics).

## Expression and purification of LeuT

Expression and purification of LeuT variants containing the tmFRET pair mutations were performed essentially as described previously (*Billesbølle et al., 2016*). In brief, *E. coli* C41(DE3) cells were transformed with pET16b plasmid encoding the desired LeuT Na1 site variants and single colonies were cultivated at 37°C in Lysogeny Broth. Expression was induced at $OD_{600}$~0.6 upon induction with β-D-1-thiogalactopyranoside (IPTG). Harvested cells were disrupted using a cell disruptor (CF1, ConstantSystems) and isolated crude membranes were solubilized with 1.5% (wt/vol) DDM (anagrade, Anatrace). Solubilized LeuT was incubated with $Ni^{2+}$-NTA resin (Thermo Fisher Scientific) and 40 mM imidazole, batch-washed and labeled overnight with F5M (Thermo Fisher Scientific). Following wash of the resin with 15 successive column volumes of buffer (20 mM Tris-HCl [pH 7.4], 20% [vol/vol] glycerol, 200 mM KCl, 0.05% [wt/vol] DDM, 100 µM tris(2-carboxyethyl)phosphine [TCEP] [hydrochloride solution]) containing 90 mM imidazole, immobilized LeuT was eluted and frozen in buffer with 340 mM

imidazole. Labeling efficiency and specificity were examined by absorbance measurements (280 and 490 nm) and sodium-dodecyl-sulfate polyacrylamide gel electrophoresis (SDS-PAGE) analysis, respectively. For LeuT variants on WT background (devoid of tmFRET mutations), the F5M labeling step was skipped and the resin washing procedure was performed with 3 and 5 column volumes of buffer containing 60 and 90 mM imidazole, respectively. For reconstitution in liposomes, LeuT was dialyzed overnight in buffer (20 mM Tris-HCl [pH 7.4], 20% [vol/vol] glycerol, 200 mM KCl, 0.05% [wt/vol] DDM) to remove imidazole. The protein was subsequently concentrated to >2 mg/ml.

## Pharmacological characterization of LeuT mutants

Pharmacological studies were conducted on purified LeuT variants by virtue of the SPA. Leucine affinities for WT LeuT and mutants displaying WT substrate affinities (A22V, A22S) were determined by saturation binding of [³H]leucine (25 Ci/mmol) (PerkinElmer), with unspecific binding corrected for by the addition of 100 µM unlabeled leucine. Alanine and leucine affinities for the remaining LeuT variants were determined by the ability of increasing concentrations of unlabeled alanine or leucine, respectively, to competitively displace a fixed concentration of [³H]leucine. Assayed in a 96-well plate (Corning), LeuT was mixed with Yttrium Silicate Copper (YSi-Cu) His-tag SPA beads (PerkinElmer) and [³H]leucine in binding buffer (200 mM NaCl, 20 mM Tris [pH 8], 20% [vol/vol] glycerol, 0.05% [wt/vol] DDM, and 100 µM TCEP). The following conditions were applied for the mutants tested: WT, K398C, A22V, A22S: 0.3 µg/ml protein, 1.25 mg/ml YSi-Cu His-tag SPA beads, 120 nM [³H]leucine (25 Ci/mmol); N27Q, N286Q: 3 µg/ml protein, 1.6 mg/ml YSi-Cu His-tag SPA beads, 1200 nM [³H]leucine (4 Ci/mmol); E290Q: 1.5 µg/ml protein, 1.6 mg/ml YSi-Cu His-tag SPA beads, 200 nM [³H]leucine (20 Ci/mmol). The Na⁺ dependency on [³H]leucine binding was determined by mixing LeuT, YSi-Cu His-tag SPA beads (in equivalent concentrations as above) and a fixed concentration of [³H]leucine (10× $K_d$, as determined in 200 mM NaCl) in binding buffer supplemented with increasing concentrations of Na⁺. The specific activity of [³H]leucine was kept approximately inversely proportional to the concentration of [³H]leucine and protein used. This experiment was repeated in the presence of 200 and 800 mM K⁺. The K⁺-dependent inhibition of Na⁺-mediated [³H]leucine binding was assayed by subjecting LeuT mutants (double the concentration as above) to increasing concentrations of K⁺ (0–1600 mM) in the presence of Na⁺ equivalent to the EC₅₀ determined with 10× $K_d$ of [³H]leucine. The unspecific binding of [³H]leucine was determined upon addition of 100 (WT, A22V, A22S) or 300 µM (N27Q, N286Q) of unlabeled leucine. Ionic strengths were preserved by substituting Na⁺ and K⁺ for Ch⁺. For competition binding and ion-dependent [³H]leucine binding experiments, ligand depletion was avoided by maintaining ≥20-fold molar excess of [³H]leucine relative to LeuT. Sealed plates were incubated for ~16 hr at 4°C with gentle agitation and counts per minute (c.p.m.) were recorded on a 2450 MicroBeta² microplate counter (PerkinElmer) in 'SPA' mode.

## Analysis of SPA data

Saturation binding data were corrected for unspecific binding, normalized to $B_{max}$ and fitted to a one-site binding regression from which $K_d$ values were obtained using GraphPad Prism 9.0. For competition binding, data points were normalized to a control (without competing unlabeled leucine or alanine) and fitted to a single-site log(inhibitor)-response model. Derived IC₅₀ values (inhibitor concentration that reduces binding of radioligand by 50%) were converted to inhibition constants $K_i$ by the Cheng-Prusoff equation:

$$K_i = \frac{IC_{50}}{1 + \frac{[S]}{K_d}}$$

where [S] and $K_d$ refer to the concentration and affinity, respectively, of the radioligand. Data points for K⁺ competition binding were corrected for unspecific binding and normalized to a control (absent for K⁺), whereas data for Na⁺-dependent [³H]leucine binding were normalized to the maximum response predicted by the model. Both were fitted to the Hill equation from which IC₅₀ and EC₅₀ values, respectively, were extracted. All experiments were performed at least three independent times in triplicates for specific and triplicates/duplicates for unspecific binding. Data points and Hill slopes are reported as mean ± s.e.m. and $K_d$, $K_i$, and EC₅₀ values are reported as [s.e.m. interval]. Statistical analysis

was performed with a post hoc Dunnett's multiple test in a one-way analysis of variance (ANOVA), comparing every mean with that obtained in the absence of competitor.

## Transition metal ion FRET

LeuT Na1 site mutants examined with tmFRET were encoded with a set of FRET pair mutations (A313H-A317H-K398C/K398C) designed and characterized previously (*Billesbølle et al., 2016*). Purified and fluorescein-labeled LeuT variants were centrifuged at 15,000 × *g* at 4°C for 10 min, and diluted to ~10 nM (adjusted for labeling efficiency) in fluorescence buffer (20 mM Tris-Cl [pH 8], 0.05% [wt/vol] DDM, 100 μM TCEP), supplemented with 800 mM chloride salts of $NMDG^+$, $Na^+ \pm 50$ μM leucine/200 μM alanine, $K^+$ or $Rb^+$ as specified. Samples were incubated for 30 min at room temperature in the dark. If not otherwise specified, parallel experiments on LeuT constructs with and without the His-$X_3$-His motif were conducted, here referred to with the suffixes tmFRET and K398C, respectively. Fluorescence measurements using a single saturating concentration of $Ni^{2+}$ were assayed in a Swartz cuvette (Hellma Analytics) inserted into a FluoroMax-4 spectrophotometer (HORIBA Scientific) temperature controlled to 25°C. Reference-corrected fluorescence intensities (0.1 s integration time) were recorded following the incubation with 10 mM $Ni^{2+}$, using constant excitation and emission wavelengths of 492 and 512 nm, respectively, and 4 nm excitation and emission slit widths. For ion titration experiments, LeuT_tmFRET was incubated in fluorescence buffer containing 750 μM $Ni^{2+}$ and increasing concentrations of $Na^+$, $K^+$, or $Rb^+$ (0–1584 mM), substituted with $Ch^+$ to preserve ionic strength. Fluorescence intensities were obtained using a FluoroMax-4 spectrophotometer and the same technical specifications as described above. For $Ni^{2+}$ titration experiments performed in 96-well plates (Corning), LeuT was subjected to increasing concentrations of $Ni^{2+}$ ($10^{-7}$ to $10^{-2}$ M) in buffer containing 800 mM of $NMDG^+$, $Na^+ \pm 50$ μM leucine, or $K \pm 50$ μM leucine. Fluorescence intensities at 512 nm were recorded on a PolarStar Omega plate reader (NMG Labtech) upon excitation at 492 nm.

## Analysis of tmFRET data

Fluorescence intensities for LeuT_tmFRET variants (*F*) were normalized to their equivalents for LeuT_K398C ($F_{no site}$) without the metal-chelating His-$X_3$-His site, to correct for the contribution of collisional quenching from free $Ni^{2+}$, dilution, as well as the primary inner filter effect ($1 - F/F_{no site}$). An increase in $1 - F/F_{no site}$ implies an enhanced energy transfer between FRET donor (F5M) and acceptor ($Ni^{2+}$) that, when plotted as a function of increasing $Ni^{2+}$, was fitted to a Hill equation from which $EC_{50}$ and maximal tmFRET values were obtained. Maximal tmFRET efficiencies were converted to distances by the FRET equation:

$$E = \frac{1}{1 + \left(\dfrac{R}{R_0}\right)^6}$$

with *E*, *R,* and $R_0$ being the efficiency of energy transfer, distance between FRET probes, and Förster distance ($R_0$=12 Å for $Ni^{2+}$/fluorescein; *Taraska et al., 2009b*), respectively. For ion titration experiments, fluorescence intensities for LeuT_tmFRET (not normalized to LeuT_K398C) at each ion concentration (*F*) were normalized to fluorescence intensities obtained in the absence ($F_0$) of $Na^+$, $K^+$, or $Rb^+$ ($F_0/F$). As for $1 - F/F_{no site}$, $F_0/F$ is inversely dependent on the distance between the FRET probes and when plotted against the ion concentration, data points could be fitted to a Hill equation, yielding the Hill slope and $EC_{50}$ value. Experiments were repeated at least three independent times in triplicates using protein from two separate purifications. Data points and Hill slopes are reported as mean ± s.e.m., whereas $EC_{50}$ values are mean [s.e.m. interval]. When indicated, data points were subjected to statistical analysis using either post hoc Tukey or Bonferroni multiple comparison test as part of a one-way ANOVA.

## Reconstitution of LeuT in liposomes

*E. coli* polar lipid extract dissolved in chloroform (Avanti Polar Lipids) was dried under a steam of $N_2$ for 2 hr. Lipids were re-suspended to 10 mg/ml in reconstitution buffer (200 mM NMDG-Cl, 20 mM Tris/HEPES [pH 7.5]) by vortexing and 2× 10 min bath sonication. The lipid solution was subjected to five freeze-thaw cycles in ethanol dry ice bath. Subsequently, the liposomes were extruded with a mini extruder (Avanti Polar Lipids) 11 times through a Nuclepore Track-Etch Membrane polycarbonate

filter of pore size 400 nm (GE Healthcare Life Sciences) and diluted to 4 mg/ml in reconstitution buffer. Liposome destabilization was induced by stepwise addition of 10 µl aliquots of 10% (vol/vol) Triton X-100. Destabilization of liposomes was followed by measuring absorbance at 550 nm. Triton X-100 was added until absorbance of the sample had reached a maximum and started to decrease. LeuT solubilized in DDM was added in a protein to lipid ratio of 1:25 (wt/wt) and the protein-liposome solution was incubated under slow rotation for 30 min at 4°C. Semidry SM-2 Bio-Beads (Bio-Rad Laboratories), equilibrated in reconstitution buffer, were added in aliquots of 8.5 mg beads/mg lipid after 30 min, 60 min, 120 min and after overnight incubation at 4°C with gentle agitation. The beads were filtered out 2 hr after the last addition of bio-beads. The proteoliposome solution was diluted ~20 times in the indicated internal buffer (200 mM salt [NaCl, KCl, NMDG-Cl, CsCl, or RbCl], 20 mM Tris/HEPES [pH 7.5]) and centrifuged for 1 hr at $140,000 \times g$ at 4°C. Pelleted proteoliposomes were re-suspended in the indicated internal buffer to a final concentration of 10 mg lipid/ml. Single-use aliquots of proteoliposomes were flash-frozen in liquid $N_2$ and stored at –80°C until further use.

## [³H]Alanine uptake into proteoliposomes

Proteoliposomes were thawed and extruded through a Nuclepore Track-Etch Membrane polycarbonate filter with 400 nm pore size (GE Healthcare Life Sciences). Uptake was assayed in a 96-well setup in ultra-low attachment, round bottom plate (Costar) at room temperature (21–23°C). Uptake buffer (20 mM Tris/HEPES [pH 7.5], 200–225 mM salt [NaCl, KCl, or NMDG-Cl]) supplemented with [³H]alanine (Moravek Biochemicals) was added to each well in a volume of 190 µl. Subsequently, 10 µl of proteoliposome solution was added to start the uptake reaction. For time-dependent uptake, the uptake buffer was supplemented with 2 µM [³H]alanine with a specific activity of 1.335 Ci/mmol and proteoliposomes were added at the indicated time points (2–60 min). For concentration-dependent uptake, uptake buffer was supplemented with [³H]alanine with a specific activity of 3.73 Ci/mmol (50–2000 nM) or 1.07 Ci/mmol (3.5–6 µM). The uptake reaction was terminated after 5 min. To assess nonspecific [³H]alanine uptake and binding, proteoliposomes were pre-incubated with 200 µM unlabeled leucine for 15 min, and 100 µM unlabeled leucine was added to the uptake buffer to saturate all transporters with leucine. The uptake reaction was terminated by filtering the samples through a 96-well glass fiber filter (Filtermat B – GF/B, Perkin Elmer) soaked in 1.5% poly(ethyleneimine) solution, using a cell harvester (Tomtec harvester 96 match II). The filter was washed with 1 ml ice-cold wash buffer (200 mM ChCl, 20 mM Tris/HEPES [pH 7.5]) for each of the 96 positions on the filter. Filter plates were dried at 96°C before scintillation sheet (MeltiLex B/HS, Perkin Elmer) was melted on to the filter. Filters were counted on a 2450 MicroBeta² microplate counter (PerkinElmer) in 'normal' counting mode.

## SDS-PAGE analysis of LeuT orientation

As for uptake experiments, proteoliposomes were extruded through a Nuclepore Track-Etch Membrane polycarbonate filter with 400 nm pore size (GE Healthcare Life Sciences). Reconstituted LeuT corresponding to 8 µg was treated with thrombin (0.5 unit; Cytiva) and the reaction was quenched upon the addition of 500 µM PMSF after 1, 5, 10, 30, 60, 120, and 180 min. Controls without thrombin and with 1.5% DDM (for 180 min) were included. Following SDS-PAGE, the gel was stained with InstantBlue (abcam) and images were obtained using an ImageQuant 800 (Cystiva).

## Radioactive binding assay for proteoliposomes

To assess the amount of active protein in each reconstitution condition, a sample of proteoliposomes from each of the different intra-vesicular buffer conditions was solubilized in buffer (30% glycerol, 1% [wt/vol] DDM, 20 mM Tris/HEPES [pH 7.5], 200 mM NaCl). The samples were left for 3 hr with gentle agitation at 4°C to dissolve proteoliposomes and solubilize LeuT in DDM detergent micelles. The amount of re-solubilized active LeuT was assessed by binding in a saturating [³H]leucine concentration (1 µM) in buffer (200 mM NaCl, 20 mM Tris [pH 8], 0.05% [wt/vol] DDM) by SPA. Measures of maximum binding from each condition were used to normalize the c.p.m. obtained from uptake experiments to ensure that uptake was not affected by variations in protein content in the proteoliposome samples. The highest maximum binding was used as normalization standard.

## Analysis of [³H]alanine uptake data

For **time-dependent uptake**, specific uptake data, normalized to the amount of active protein (see section above), was fitted to a one-phase association using GraphPad Prism 9.0.

$$y\left(t\right) = p * e^{\left(-k*t\right)}$$

where $p$ is the maximal uptake (the amplitude), $k$ is the rate constant in min$^{-1}$, $t$ is time in min, and $y$ is uptake in c.p.m. after normalization. Data from each experiment was subsequently normalized to the value of $p$ from the condition with NMDG$^+$. The normalized data sets from each experiment were combined and re-fitted to a one-phase association.

For **concentration-dependent uptake**, normalized, specific uptake data was fitted to the Michaelis-Menten equation:

$$y = \frac{V_{max} * \left[s\right]}{K_m + \left[s\right]}$$

where $V_{max}$ is the maximal uptake rate, $K_m$ is the substrate concentration where half-maximum uptake rate is reached, $y$ is the uptake in c.p.m. after normalization. $[s]$ is the [³H]alanine concentration in µM. Subsequently data was either combined or (if shown in % on the y-axis) normalized to the $V_{max}$ value determined with intra-vesicular NMDG$^+$. The normalized data sets from each experiment were combined and fitted to the Michaelis-Menten equation. All uptake experiments were done in triplicates or duplicates as indicated and repeated three to four times using protein obtained from at least two different purifications and reconstitutions.

For concentration-dependent uptake with different intra-vesicular cations, c.p.m. were converted to d.p.m., and then to pmol of [³H]alanine per min, using the specific activity of [³H]alanine and a counting efficiency of 40% for 2450 MicroBeta$^2$ microplate counter (PerkinElmer) in 'normal' counting mode.

## Materials availability statement

All newly created materials (e.g. LeuT mutants) are available upon request.

# Acknowledgements

We would like to thank Patricia Curran for technical guidance. We also acknowledge the NINDS intramural program for support. Funding sources: Support for this research was provided by the Independent Research Fund Denmark (1030-00036B to CJL), the Lundbeck Foundation (R344-2020-1020 to CJL), the Novo Nordic Foundation (NNF19OC0058496 to CJL), and the Carlsberg Foundation (CF20-0345 to CJL).

# Additional information

### Funding

| Funder | Grant reference number | Author |
| --- | --- | --- |
| Danmarks Frie Forskningsfond | 1030-00036B | Claus J Loland |
| Lundbeck Foundation | R344-2020-1020 | Claus J Loland |
| Novo Nordisk Fonden | NNF19OC0058496 | Claus J Loland |
| Carlsbergfondet | CF20-0345 | Claus J Loland |

The funders had no role in study design, data collection and interpretation, or the decision to submit the work for publication.

### Author contributions

Solveig G Schmidt, Andreas Nygaard, Data curation, Formal analysis, Investigation, Methodology, Writing – original draft, Writing – review and editing; Joseph A Mindell, Data curation, Supervision,

**Table 1.** LeuT mutant affinities for leucine and alanine.

Affinities were determined by inhibition of [³H]leucine with either leucine or alanine. The IC$_{50}$ values, obtained by non-linear regression fit, were converted to $K_d$ and $K_i$ values by the Cheng-Prusoff equation. For [A]-marked $K_d$'s, affinities were obtained by [³H]leucine saturation binding. All experiments were assayed in 200 mM Na$^+$ and data are reported as mean [s.e.m. interval].

| | Leucine affinity $K_d$ (nM) | n | Alanine affinity $K_i$ (µM) | n |
|---|---|---|---|---|
| WT | 6.9 [6.7; 7.2][A] | 4 | 1.61 [1.58; 1.65] | 4 |
| A22V | 5.4 [4.9; 5.9][A] | 3 | 1.02 [0.97; 1.08] | 4 |
| A22S | 8.4 [7.7; 9.1][A] | 3 | 2.11 [2.05; 2.17] | 4 |
| N27Q | 1590 [1530; 1650] | 3 | 255 [248; 263] | 3 |
| T254S | 5.6 [5.4; 5.8][A] | 3 | 1.17 [1.12; 1.21] | 3 |
| N286Q | 1460 [1390; 1540] | 3 | 287 [277; 297] | 3 |
| E290Q | 193 [169; 221] | 4 | 30.0 [28.9; 31.2] | 3 |

The online version of this article includes the following source data for table 1:

**Source data 1.** Pdf file with affinity curves for leucine and alanine binding to LeuT and mutants.

**Source data 2.** Excel file containing data for leucine and alanine affinities for LeuT WT and all investigated mutants.

Funding acquisition, Methodology, Project administration, Writing – review and editing; Claus J Loland, Conceptualization, Formal analysis, Supervision, Funding acquisition, Writing – original draft, Project administration, Writing – review and editing

**Author ORCIDs**
Solveig G Schmidt http://orcid.org/0000-0002-4771-1227
Andreas Nygaard http://orcid.org/0000-0001-8651-128X
Joseph A Mindell https://orcid.org/0000-0002-6952-8247
Claus J Loland http://orcid.org/0000-0002-1773-1446

Joint Public Review: https://doi.org/10.7554/eLife.87985.3.sa1
Author Response https://doi.org/10.7554/eLife.87985.3.sa2

## Additional files

**Supplementary files**
• MDAR checklist

### Data availability

All data generated or analysed during this study are included in the manuscript and supporting file. Source data files have been provided for *Figures 1–5* and for *Tables 1 and 2*.

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

**Table 2.** Na$^+$-mediated [$^3$H]leucine binding and the effect of K$^+$ addition, and K$^+$-dependent Na$^+$/[$^3$H]leucine displacement in Na1 site mutants.

For LeuT Na1 site mutants (and K398C), Na$^+$-mediated [$^3$H]leucine binding was assayed using 10× $K_d$ of [$^3$H]leucine in the presence of 0, 200, and 800 mM K$^+$. For K$^+$-dependent displacement of Na$^+$-mediated [$^3$H]leucine binding, 10× $K_d$ of [$^3$H]leucine and Na$^+$ equivalent to the EC$_{50}$ value for each of the mutants were used. Na$^+$ and K$^+$ were substituted for Ch$^+$ to preserve the ionic strength. Data were fitted to a Hill equation, yielding EC$_{50}$ and IC$_{50}$ values reported here as mean [s.e.m. interval], $n$=3–6 determined in triplicates. Comparing the Na$^+$ EC$_{50}$ obtained for the mutants with that of WT showed significant difference for all (p<0.0001) except for E290Q (p>0.05), and for the K$^+$ IC$_{50}$ all showed significant differences from WT (p<0.0001) expect for T254S (p>0.05) (Dunnett's multiple comparison-corrected one-way analysis of variance [ANOVA]). N.d. indicates that the value is not determined. WT data are also reported in *Figure 1*.

| | EC$_{50}$ Na$^+$ (mM) | IC$_{50}$ K$^+$ (mM) | EC$_{50}$ Na$^+$+200 mM K$^+$ (mM) | EC$_{50}$ Na$^+$+800 mM K$^+$ (mM) | EC$_{50}$$^{800}$/EC$_{50}$ |
|---|---|---|---|---|---|
| WT | 7.7 [7.3; 8.1] | 235 [228; 241] | 19.4 [19.1; 19.8] | 48.5 [47.5; 49.5] | ~6 |
| A22V | 2.0 [2.0; 2.1] | 49.0 [47.7; 50.4] | 23.0 [21.7; 24.5] | 70.7 [69.2; 72.2] | ~35 |
| A22S | 53.3 [45.5; 62.5] | >1900 | 72.4 [62.3; 84.2] | 117 [100; 137] | ~2 |
| N27Q | 2.4 [2.3; 2.5] | >2,300 | n.d. | 4.4 [4.0; 4.8] | ~2 |
| T254S | 15.1 [14.2; 16.0] | 219 [209; 229] | 43.5 [37.4; 50.5] | 90.2 [73.8; 110] | ~6 |
| N286Q | 54.6 [47.5; 62.6] | >2000 | n.d. | 121 [111; 133] | ~2 |
| E290Q | 9.6 [8.8; 10.4] | n.d. | n.d. | 144 [132; 157] | ~15 |
| K398C | 7.7 [6.8; 8.8] | n.d | n.d. | 51.5 [48.1; 55.1] | ~7 |

Calugareanu D, Möller IR, Schmidt SG, Loland CJ, Rand KD. 2022. Probing the impact of temperature and substrates on the conformational dynamics of the neurotransmitter:sodium symporter LeuT. *Journal of Molecular Biology* **434**:167356. DOI: https://doi.org/10.1016/j.jmb.2021.167356, PMID: 34780780

Di Giovanni G, Svob Strac D, Sole M, Unzeta M, Tipton KF, Mück-Šeler D, Bolea I, Della Corte L, Nikolac Perkovic M, Pivac N, Smolders IJ, Stasiak A, Fogel WA, De Deurwaerdère P. 2016. Monoaminergic and histaminergic strategies and treatments in brain diseases. *Frontiers in Neuroscience* **10**:541. DOI: https://doi.org/10.3389/fnins.2016.00541, PMID: 27932945

Fitzgerald GA, Terry DS, Warren AL, Quick M, Javitch JA, Blanchard SC. 2019. Quantifying secondary transport at single-molecule resolution. *Nature* **575**:528–534. DOI: https://doi.org/10.1038/s41586-019-1747-5, PMID: 31723269

Forrest LR, Tavoulari S, Zhang YW, Rudnick G, Honig B. 2007. Identification of a chloride ion binding site in Na+/Cl -dependent transporters. *PNAS* **104**:12761–12766. DOI: https://doi.org/10.1073/pnas.0705600104, PMID: 17652169

Hasenhuetl PS, Freissmuth M, Sandtner W. 2016. Electrogenic binding of intracellular cations defines a kinetic decision point in the transport cycle of the human serotonin transporter. *The Journal of Biological Chemistry* **291**:25864–25876. DOI: https://doi.org/10.1074/jbc.M116.753319, PMID: 27756841

Jayanthi LD, Ramamoorthy S. 2005. Regulation of monoamine transporters: influence of psychostimulants and therapeutic antidepressants. *The AAPS Journal* **7**:E728–E738. DOI: https://doi.org/10.1208/aapsj070373, PMID: 16353949

Kantcheva AK, Quick M, Shi L, Winther A-ML, Stolzenberg S, Weinstein H, Javitch JA, Nissen P. 2013. Chloride binding site of neurotransmitter sodium symporters. *PNAS* **110**:8489–8494. DOI: https://doi.org/10.1073/pnas.1221279110, PMID: 23641004

Keyes SR, Rudnick G. 1982. Coupling of transmembrane proton gradients to platelet serotonin transport. *The Journal of Biological Chemistry* **257**:1172–1176 PMID: 7056713.

Khelashvili G, Schmidt SG, Shi L, Javitch JA, Gether U, Loland CJ, Weinstein H. 2016. Conformational dynamics on the extracellular side of LeuT controlled by Na+ and K+ ions and the protonation state of Glu290. *The Journal of Biological Chemistry* **291**:19786–19799. DOI: https://doi.org/10.1074/jbc.M116.731455, PMID: 27474737

Krishnamurthy H, Gouaux E. 2012. X-ray structures of LeuT in substrate-free outward-open and apo inward-open states. *Nature* **481**:469–474. DOI: https://doi.org/10.1038/nature10737, PMID: 22230955

Kristensen AS, Andersen J, Jørgensen TN, Sørensen L, Eriksen J, Loland CJ, Strømgaard K, Gether U. 2011. SLC6 neurotransmitter transporters: structure, function, and regulation. *Pharmacological Reviews* **63**:585–640. DOI: https://doi.org/10.1124/pr.108.000869, PMID: 21752877

Lakowicz JR. 2006. *Principles of Fluorescence Spectroscopy* Springer. DOI: https://doi.org/10.1007/978-0-387-46312-4

Li LB, Reith MEA. 2000. Interaction of Na+, K+, and Cl- with the binding of amphetamine, octopamine, and tyramine to the human dopamine transporter. *Journal of Neurochemistry* **74**:1538–1552. DOI: https://doi.org/10.1046/j.1471-4159.2000.0741538.x, PMID: 10737611

Loland CJ. 2015. The use of LeuT as a model in elucidating binding sites for substrates and inhibitors in neurotransmitter transporters. *Biochimica et Biophysica Acta* **1850**:500–510. DOI: https://doi.org/10.1016/j.bbagen.2014.04.011, PMID: 24769398

Malinauskaite L, Said S, Sahin C, Grouleff J, Shahsavar A, Bjerregaard H, Noer P, Severinsen K, Boesen T, Schiøtt B, Sinning S, Nissen P. 2016. A conserved leucine occupies the empty substrate site of LeuT in the Na(+)-free return state. *Nature Communications* **7**:11673. DOI: https://doi.org/10.1038/ncomms11673, PMID: 27221344

Mari SA, Soragna A, Castagna M, Bossi E, Peres A, Sacchi VF. 2004. Aspartate 338 contributes to the cationic specificity and to driver-amino acid coupling in the insect cotransporter KAAT1. *Cellular and Molecular Life Sciences* **61**:243–256. DOI: https://doi.org/10.1007/s00018-003-3367-2, PMID: 14745502

Möller IR, Slivacka M, Nielsen AK, Rasmussen SGF, Gether U, Loland CJ, Rand KD. 2019. Conformational dynamics of the human serotonin transporter during substrate and drug binding. *Nature Communications* **10**:1687. DOI: https://doi.org/10.1038/s41467-019-09675-z, PMID: 30976000

Motiwala Z, Aduri NG, Shaye H, Han GW, Lam JH, Katritch V, Cherezov V, Gati C. 2022. Structural basis of GABA reuptake inhibition. *Nature* **606**:820–826. DOI: https://doi.org/10.1038/s41586-022-04814-x, PMID: 35676483

Nelson PJ, Rudnick G. 1979. Coupling between platelet 5-hydroxytryptamine and potassium transport. *The Journal of Biological Chemistry* **254**:10084–10089 PMID: 489585.

Quick M, Javitch JA. 2007. Monitoring the function of membrane transport proteins in detergent-solubilized form. *PNAS* **104**:3603–3608. DOI: https://doi.org/10.1073/pnas.0609573104, PMID: 17360689

Rudnick G, Nelson PJ. 1978. Platelet 5-hydroxytryptamine transport, an electroneutral mechanism coupled to potassium. *Biochemistry* **17**:4739–4742. DOI: https://doi.org/10.1021/bi00615a021, PMID: 728383

Ryan RM, Compton ELR, Mindell JA. 2009. Functional characterization of a Na+-dependent aspartate transporter from Pyrococcus horikoshii. *The Journal of Biological Chemistry* **284**:17540–17548. DOI: https://doi.org/10.1074/jbc.M109.005926, PMID: 19380583

Schicker K, Uzelac Z, Gesmonde J, Bulling S, Stockner T, Freissmuth M, Boehm S, Rudnick G, Sitte HH, Sandtner W. 2012. Unifying concept of serotonin transporter-associated currents. *The Journal of Biological Chemistry* **287**:438–445. DOI: https://doi.org/10.1074/jbc.M111.304261, PMID: 22072712

Schmidt SG, Malle MG, Nielsen AK, Bohr SSR, Pugh CF, Nielsen JC, Poulsen IH, Rand KD, Hatzakis NS, Loland CJ. 2022. The dopamine transporter antiports potassium to increase the uptake of dopamine. *Nature Communications* **13**:2446. DOI: https://doi.org/10.1038/s41467-022-30154-5, PMID: 35508541

Shahsavar A, Stohler P, Bourenkov G, Zimmermann I, Siegrist M, Guba W, Pinard E, Sinning S, Seeger MA, Schneider TR, Dawson RJP, Nissen P. 2021. Structural insights into the inhibition of glycine reuptake. *Nature* **591**:677–681. DOI: https://doi.org/10.1038/s41586-021-03274-z, PMID: 33658720

Shi L, Quick M, Zhao Y, Weinstein H, Javitch JA. 2008. The mechanism of a neurotransmitter:sodium symporter--inward release of Na+ and substrate is triggered by substrate in a second binding site. *Molecular Cell* **30**:667–677. DOI: https://doi.org/10.1016/j.molcel.2008.05.008, PMID: 18570870

Taraska JW, Puljung MC, Olivier NB, Flynn GE, Zagotta WN. 2009a. Mapping the structure and conformational movements of proteins with transition metal ion FRET. *Nature Methods* **6**:532–537. DOI: https://doi.org/10.1038/nmeth.1341, PMID: 19525958

Taraska JW, Puljung MC, Zagotta WN. 2009b. Short-distance probes for protein backbone structure based on energy transfer between bimane and transition metal ions. *PNAS* **106**:16227–16232. DOI: https://doi.org/10.1073/pnas.0905207106, PMID: 19805285

Wang KH, Penmatsa A, Gouaux E. 2015. Neurotransmitter and psychostimulant recognition by the dopamine transporter. *Nature* **521**:322–327. DOI: https://doi.org/10.1038/nature14431, PMID: 25970245

Wang J, Zielewicz L, Grewer C. 2019. A K+/Na+ co-binding state: Simultaneous versus competitive binding of K+ and Na+ to glutamate transporters. *Journal of Biological Chemistry* **294**:12180–12190. DOI: https://doi.org/10.1074/jbc.RA119.009421

Yamashita A, Singh SK, Kawate T, Jin Y, Gouaux E. 2005. Crystal structure of a bacterial homologue of Na+/Cl--dependent neurotransmitter transporters. *Nature* **437**:215–223. DOI: https://doi.org/10.1038/nature03978, PMID: 16041361

Yang D, Gouaux E. 2021. Illumination of serotonin transporter mechanism and role of the allosteric site. *Science Advances* **7**:eabl3857. DOI: https://doi.org/10.1126/sciadv.abl3857, PMID: 34851672

Zhao Y, Terry DS, Shi L, Quick M, Weinstein H, Blanchard SC, Javitch JA. 2011. Substrate-modulated gating dynamics in a Na+-coupled neurotransmitter transporter homologue. *Nature* **474**:109–113. DOI: https://doi.org/10.1038/nature09971, PMID: 21516104

Zomot E, Bendahan A, Quick M, Zhao Y, Javitch JA, Kanner BI. 2007. Mechanism of chloride interaction with neurotransmitter:sodium symporters. *Nature* **449**:726–730. DOI: https://doi.org/10.1038/nature06133, PMID: 17704762

