## [Editor Report · eLife assessment]

The bacterial neurotransmitter:sodium symporter homoglogue LeuT is a well-established model system for understanding the **fundamental** basis for how human monoamine transporters, such as those for dopamine and serotonin, couple ions with neurotransmitter uptake. Here the authors provide **convincing** data to show that K^+^ binding on the intraceullular side catalyses the return step of the transport cycle in LeuT by binding to one of the two sodium sites. The mechansitic consequences of K^+^ binding could either facilitate LeuT re-setting and/or prevent the rebinding and possible efflux of Na^+^ and substrate.

---

## [Referee Report · Joint Public Review]

This manuscript tackles an important question, namely how K+ affects substrate transport in the SLC6 family. K+ effects have previously been reported for DAT and SERT, but the prototypical SLC6-fold transporter LeuT was not known to be sensitive to the K+ concentration. In this manuscript, the authors demonstrate convincingly that K+ inhibits Na+ binding, and Na+-dependent amino acid binding at high concentrations, and that K+ inside of vesicles containing LeuT increases the transport rate. However, outside K+ apparently had very little effect. Uptake data are supplemented with binding data, using the scintillation proximity assay, and transition metal FRET, allowing the observation of the distribution of distinct conformational states of the transporter.

Overall, the data are of high quality. I was initially concerned about the use of solutions of very high ionic strength (the Km for K+ is in the 200 mM range), however, the authors performed good controls with lower ionic strength solutions, suggesting that the K+ effect are specific and not caused by artifacts from the high salt concentrations.

---

## [Author Response]

The following is the authors’ response to the original reviews.

**eLife assessment**
The bacterial neurotransmitter:sodium symporter homoglogue LeuT is an well-established model system for understanding the fundamental basis for how human monoamine transporters, such as the dopamine and serotonin, couple ions with neurotransmitter uptake. Here the authors provide convincing data to show that the K+ catalyses the return step of the transport cycle in LeuT by binding to one of the two sodium sites. The paper is an important contribution, but it's still unclear exactly where K+ binds in LeuT, and how to incorporate K+ binding into a transport cycle mechanism.
**Public Reviews:**

**Reviewer #1 (Public Review):**
This manuscript tackles an important question, namely how K+ affects substrate transport in the SLC6 family. K+ effects have previously been reported for DAT and SERT, but the prototypical SLC6fold transporter LeuT was not known to be sensitive to the K+ concentration. In this manuscript, the authors demonstrate convincingly that K+ inhibits Na+ binding, and Na+-dependent amino acid binding at high concentrations, and that K+ inside of vesicles containing LeuT increases the transport rate. However, outside K+ apparently had very little effect. Uptake data are supplemented with binding data, using the scintillation proximity assay, and transition metal FRET, allowing the observation of the distribution of distinct conformational states of the transporter.Overall, the data are of high quality. I was initially concerned about the use of solutions of very high ionic strength (the Km for K+ is in the 200 mM range), however, the authors performed good controls with lower ionic strength solutions, suggesting that the K+ effect is specific and not caused by artifacts from the high salt concentrations.The major issue I have with this manuscript is with the interpretation of the experimental data. Granted that the K+ effect seems to be complex. However, it seems counterintuitive that K+ competes with Na+ for the same binding site, while at the same time accelerating the transport rate. Even if K+ prevents rebinding of Na+ on the inside of vesicles, it would be expected that K+ then stabilizes this Na+-free conformation, resulting in a slowing of the transport rate. However, the opposite is found. I feel that it would be useful to perform some kinetic modeling of the transport cycle to identify a mechanism that would allow K+ to act as a competitive inhibitor of Na+ binding and rate-accelerator at the same time.This ties into the second point: It is not mentioned in the manuscript what the configuration of the vesicles is after LeuT reconstitution. Are they right-side out? Is LeuT distributed evenly in inside-out and right-side out orientation? Is the distribution known? If yes, how does it affect the interpretation of the uptake data with and without K+ gradient?Finally, mutations were only made to the Na1 cation binding site. These mutations have an effect mostly to be expected, if K+ would bind to this site. However, indirect effects of mutations can never be excluded, and the authors acknowledge this in the discussion section. It would be interesting to see the effect of K+ on a couple of mutants that are far away from Na+/substrate binding sites. This could be another piece of evidence to exclude indirect effects, if the K+ affinity is less affected.
**Reviewer #2(Public Review):**
To characterize the relationship between Na+ and K+ binding to LeuT, the effect of K+ on Na+- dependent [3 H] leucine binding was studied using a scintillation proximity assay. In the presence of K+ the apparent affinity for sodium was reduced but the maximal binding capacity for this ion was unchanged, consistent with a competitive mechanism of inhibition between Na+ and K+.To obtain a more direct readout of K+ binding to LeuT, tmFRET was used. This method relies on the distance-dependent quenching of a cysteine-conjugated fluorophore (FRET donor) by a transition metal (FRET acceptor). This method is a conformational readout for both ion- and ligand-binding. Along with the effect of K+ on Na+-dependent [3 H] leucine binding, the findings support the existence of a specific K+ binding site in LeuT and that K+ binding to this site induces an outward closed conformation.It was previously shown that in liposomes inlaid with LeuT by reconstitution, intra-vesicular K+ increases the concentrative capacity of [ 3 H] alanine. To obtain insights into the mechanistic basis of this phenomenon, purified LeuT was reconstituted into liposomes containing a variety of cations, including Na+ and K+ followed by measurements of [ 3 H] alanine uptake driven by a Na+ gradient.The ionic composition of the external medium was manipulated to determine if the stimulation of [3 H] alanine uptake by K+ was due to an outward directed potassium gradient serving as a driving force for sodium-dependent substrate transport by moving in the direction opposite to that of sodium and the substrate. Remarkably it was found that it is the intra-liposomal K+ per se that increases the transport rate of alanine and not a K+ gradient, suggesting that binding of K+ to the intra-cellular face of the transporter could prevent the rebinding of sodium and the substrate thereby reducing their efflux from the cell. These conclusions assume that the measured radioactive transport is via right-side-out liposomes rather than from their inverted counterparts (in case of a random orientation of the transporters in the proteoliposomes). Even though this assumption is likely to be correct, it should be tested.Since K+- and Na+-binding are competitive and K+ excludes substrate binding, the Authors chose to focus on the Na1 site where the carboxyl group of the substrate serves as one of the groups which coordinate the sodium ion. This was done by the introduction of conservative mutations of the amino acid residues forming the Na1 site. The potassium interaction in these mutants was monitored by sodium dependent radioactive leucine binding. Moreover, the effect the effect of Na+ with and without substrate as well as that of potassium on the conformational equilibria was measured by tmFRET measurements on the mutants introduced in the construct enabling the measurements. The results suggest that K+-binding to LeuT modulates substrate transport and that the K+ affinity and selectivity for LeuT is sensitive to mutations in the Na1 site, pointing toward the Na1 site as a candidate site for facilitating the interaction between K+ in some NSS members.The data presented in this manuscript are of very high quality. They are a detailed extension of results by the same group (Billesbolle et. al, Ref. 16 from the list) providing more detailed information on the importance of the Na1 site for potassium interaction. Clearly this begs for the identification of the binding site in a potassium bound LeuT structure in the future. Presumably LeuT was studied here because it appears that it is relatively easy to determine structures of many conformational states. Furthermore, convincing evidence showed that the stimulatory effect of K+ on transport is not because of energization of substrate accumulation but is rather due to the binding of this cation to a specific site.
**Reviewer #1 (Recommendations For The Authors):**
Include a transport mechanism that can account for the K+ effects.

We appreciate the opportunity to elaborate further regarding how we envision this complex mechanism. It is generally known that, within the LeuT-fold transporters, the return step is ratelimiting for the transport process. Our data suggests that K+ binds to the inward-facing apo form.

Accordingly, we propose that the role of K+ binding is to facilitate LeuT to overcome the rate-limiting step. We propose the following mechanistic model: When Na+ and substrate is released to the intracellular environment the transporter must return to the outward-facing conformation. This can happen in (at least) two ways: (1) The transporter in its apo-form closes the inner gate and opens to the extracellular side, now ready to perform a new transport cycle. (2) The transporter rebinds Na+, which allows for the rebinding of substrate. It can now go in reverse (efflux) or it once again release its content. The transporter can naturally also only rebind Na+ and release it again to the cytosol.

The purpose of K+ binding is to prevent Na+ rebinding and to promote a conformational state of the transporter, which does not allow Na+ binding. Even though Na+ has a higher affinity for the site, K+ is much more abundant.

This model is supported by our previous experiment, showing that intravesicular K+ prevents [3H]alanine efflux while LeuT performs Na+-dependent alanine transport. Thus, the increase in Vmax could be due to a decreased efflux (exchange mode), or a facilitation of the rate-limiting step, or a combination of the two.

Note that the model does not require that K+ is counter-transported. It just has to prevent Na+ rebinding. However, even though we failed to show K+ counter-transport, it does not mean that it does not happen. Further experiments must clarify this issue.

To be more explicit about our proposed mechanistic model, we have expanded the last paragraph in the Discussion section. It now reads:

“We propose that K+ binding either facilitates LeuT transition from inward- to outward-facing (the rate limiting step of the transport cycle), or solely prevents the rebinding and possible efflux of Na+ and substrate. It could also be a combination of both. Either way, intracellular K+ will lead to an increase in Vmax and concentrative capacity. Note that our previous experiment showed an increased [3H]alanine efflux when LeuT transports alanine in the absence of intra-vesicular K+16. Specifically, the mechanistic impact of K+ could be to catalyze LeuT away from the state that allows the rebinding of Na+ and substrate. This way, K+ binding would decrease the possible rebinding of intracellularly released Na+ and substrate, thereby rectifying the transport process and increase the concentrative capacity and Vmax (Figure 6). Our results suggest that K+ is not counter-transported but rather promotes LeuT to overcome an internal rate limiting energy barrier. However, further investigations must be performed before any conclusive statement can be made here.”

Describe the orientation of the transporter in the vesicles.

When working with reconstituted NSS, the transport activity is determined by the Na+ gradient. This is also evident in the experiments where we dissipate the Na+ gradient. Here we find transport activity compatible to background. We can also see in the literature, that directionality is rarely determined for transport proteins in reconstituted systems. When that is said, it is difficult to know how the inside-out LeuT contribute to the transport process. Will they work in reverse and contribute to the accumulation of intravesicular [3H]alanine? If so, to what extent? They will likely not be affected by the intravesicular K+. Therefore, their possible contribution will ‘work against’ our results and decrease the apparent K+ effects reported herein. Taken together, unless the vast majority of LeuT molecules are inside-out, knowing the actual proportion will not, in our perspective, affect our interpretations and conclusions of the data.

When that is said, we have also been curious about this issue and with the question raised by the reviewer, we performed the suggested experiment. We have inserted the results in Figure 3 – Figure supplement 1D. The figure shows that a fraction of the reconstituted LeuT are susceptible to thrombin cleavage of the accessible C-terminal. We have quantified the cleaved fraction to around 40% of the total (see Author response image 1 below). It is, however, a crude estimate since it is difficult to perform reliable dosimetry with fractions that close together. Thus, we are reluctant to add a quantitative measure in the article text.

**Author response image 1. sa2fig1:** 

We have inserted the following in the main text:

“It is difficult to control the directionality of proteins when they are reconstituted into lipid vesicles. They will be inserted in both orientations. Outside-out and inside-out. In the case of LeuT it is the imposed Na+-gradient which is determines the directionality of transport. Uptake through the insideout transporters will probably also happen. Note that the inside-out LeuT will not have the K+ binding site exposed to the intra-vesicular environment.Accordingly, a propensity of transporters will likely not be influenced by the added K+ and will tend to mask the contribution of K+ to the transport mode from the right-side out LeuT. To investigate LeuT directionality in our reconstituted samples, we performed thrombin cleavage of accessible C-terminals on intact and perforated vesicles, respectively. The result suggests that the proportion of LeuT inserted as outside-out is larger than the proportion with an inside-out directionality (Figure 3 – Figure supplement 1D).”

For the inserted Figure 3 – Figure supplement 1D, we have added the following legend:

“(D) SDS-PAGE analysis of LeuT proteoliposomes following time-dependent thrombin digestion of accessible C-terminals (reducing the mass of LeuT by ~1.3 kDa). The reaction was terminated by the addition of PMSF at the specified time points. The lanes corresponding to the time-dependent proteolysis are flanked by lanes containing proteoliposomes without thrombin (left, 0 min) or digested in the presence of DDM (right, 180 min+DDM). Arrows indicate bands of full-length (top) and cleaved (bottom) LeuT.”

Check the effects of mutations away from the Na1 cation binding site.

We have included the LeuT K398C in the study as a negative control for unspecific effects on Na+ and K+ binding. The mutant exhibit Na+ dependent [3H]leucine binding and K+-dependency similar to LeuT WT – see Table 2 and Table 2 - Figure Supplement 1G.

As a minor point, the authors use the term "affinity" liberally. However, unless these are direct binding experiments, the term "apparent affinity" may be more appropriate, since Km values are affected by the transport cycle (in uptake), as well as binding of cations/substrate.

We thank the reviewer for emphasizing this important point. We have revised the manuscript accordingly. We use ‘affinity’ when it has been determined under equilibrium conditions, either as a SPA binding experiment or based on tmFRET. We use the term ‘Km’ when the apparent affinity has been determined during non-equilibrium conditions such as during substrate transport.

**Reviewer #2 (Recommendations For The Authors):**
As mentioned in part 2, it is important to show the effect of internal potassium on transport in-sided liposomes. This could be done using the methodology developed by Tsai et. al. Biochemistry 51 (2012) 1557-1585.

We appreciate this important point and have performed the suggested experiment. See reviewer 1 comment #2

In the Abstract and throughout it is mentioned that K+ is not counter transported, yet on the bottom of p. 16 it is mentioned that this is possible.

We have tried to be very cautious with any interpretation about whether K+ is only binding or whether it is also counter-transported. Either way, it must facilitate a transition towards a non-Na+ binding state. We tried to differentiate between the two possibilities by investigating if an outwarddirected K+ gradient alone could drive transport (Figure 3E). We do not observe any significant difference from background (no gradient). However, the gained information is rather weak: It is still possible that K+ is counter-transported, but the K+ gradient does not impose any driving force. Instead, it ensures a rectification of the Na+-dependent substrate transport. If so, this experiment would come up negative even if K+ is counter-transported.

To be more explicit, we have changed the wording on page 16.

Our results suggests that K+ is not counter-transported, but rather promote LeuT to overcome an internal rate limiting energy barrier. However, further investigations must be performed before any conclusive statement can be made here.

Fig.2-Fig. Supplement 1: it is important to show that the effect of leucine is sodium-dependent by adding the control K+ and leucine.

We thank the reviewer for suggesting this important control. We have added the experiment to Figure 2 – Figure supplement 1 as suggested. The effect is not different from K+ alone supporting the SPA-binding data that K+-binding does not promote substrate binding.

Point for discussion: Whereas potassium is counter transported in SERT, there are conflicting interpretations on this in DAT (Ref. 15 from the list and Bhat et. al eLife (2021) 10:e67996). The situation in LeuT seems like the scenario described by Bhat et. al.

We appreciate the suggestion for a proposed link between LeuT and hDAT. Although, as mentioned above, we find it early days to be too certain on this option. We have now mentioned the mechanistic similarity in the Discussion following our description of the proposed mechanistic model (see first request from reviewer #1):

“If K+ is not counter-transported, LeuT might comply with the mechanism previously suggested for the human DAT31.”

Fig. 5-Fig. Supplement 1: Why are no data on N27Q and N286Q given? If these mutants have no transport activity this should be stated. Moreover, alanine uptake by A22V is almost sodium independent and is also very fast, suggesting binding, not transport. Are the counts sensitive to ionophores like nigericin?

We appreciate this important point. Indeed, the LeuT N27Q and N286Q are transport inactive. This information is now inserted in the main text when describing the conformational dynamics of N27QtmFRET and N286QtmFRET.

We agree with the reviewer that the [3H]alanine uptake for A22V is not very conclusive. The vesicles with Na+ on both sides (open diamonds) do allow [3H]alanine binding. Vesicles with added gramicidin are similar in activity. The fast rate could indeed suggest a binding event. This we also do not rule out in the main text. However, the contribution in activity from LeuT A22V in vesicles with a Na+ gradient cannot be explained by a binding event alone. Then it should bind more [3H]alanine in the presence of a Na+ gradient, which is possible, but hard to imagine. Also, the alanine affinity for LeuT A22V is ~1 µM (Table 1). At this affinity it should be literally impossible to detect any binding because the off-rate is so fast that it would all dissociate during the washing procedure.

We have described the data and left out any interpretation (e.g. changed ‘[3H]alanine transport’ to ‘[3H]alanine activity’). In addition, we have replaced: “This correlates with the lack of changes in conformational equilibrium observed in the tmFRET data between the NMDG+, Na+ and K+ states.” with: “Further investigations must clarify whether the changes in observed [3H]alanine activity constitutes a transport- or a binding event.”

Lower part of p. 16. The Authors speculate "that the mechanistic impact of K+ binding could be to accelerate a transition away from the conformation where Na+ and substrate are released, to a state where they can no longer rebind and thus revert the transport process (efflux)". This could be easily tested by measuring exchange, which should not be influenced by potassium.

We performed this experiment in Billesbolle et al. 2016. Nat Commun (Fig. 1f). We show that the exchange is decreased in the presence of K+. We hypothesize that this is because K+ binding forces LeuT away from the exchange mode.